# Ketamine increases activity of a fronto-striatal projection that regulates compulsive behavior in SAPAP3 knockout mice

Gwynne L. Davis[1], Adelaide R. Minerva[1], Argentina Lario[1], Linda D. Simmler[2], Carolyn I. Rodriguez[3,4] & Lisa A. Gunaydin [1,5,6 ✉]

Obsessive-Compulsive Disorder (OCD), characterized by intrusive thoughts (obsessions) and repetitive behaviors (compulsions), is associated with dysfunction in fronto-striatal circuits. There are currently no fast-acting pharmacological treatments for OCD. However, recent clinical studies demonstrated that an intravenous infusion of ketamine rapidly reduces OCD symptoms. To probe mechanisms underlying ketamine's therapeutic effect on OCD-like behaviors, we used the SAPAP3 knockout (KO) mouse model of compulsive grooming. Here we recapitulate the fast-acting therapeutic effect of ketamine on compulsive behavior, and show that ketamine increases activity of dorsomedial prefrontal neurons projecting to the dorsomedial striatum in KO mice. Optogenetically mimicking this increase in fronto-striatal activity reduced compulsive grooming behavior in KO mice. Conversely, inhibiting this circuit in wild-type mice increased grooming. Finally, we demonstrate that ketamine blocks the exacerbation of grooming in KO mice caused by optogenetically inhibiting fronto-striatal activity. These studies demonstrate that ketamine increases activity in a fronto-striatal circuit that causally controls compulsive grooming behavior, suggesting this circuit may be important for ketamine's therapeutic effects in OCD.

[1] Institute for Neurodegenerative Diseases, University of California San Francisco, San Francisco, CA, USA. [2] Department of Basic Neurosciences, University of Geneva, Rue Michel-Servet 1, 1206 Geneva, Switzerland. [3] Department of Psychiatry and Behavioral Sciences, Stanford University, Stanford, CA, USA. [4] Veterans Affairs Palo Alto Health Care System, Palo Alto, CA, USA. [5] Department of Psychiatry and Behavioral Sciences, University of California San Francisco, San Francisco, CA, USA. [6] Kavli Institute for Fundamental Neuroscience, University of California San Francisco, San Francisco, CA, USA. ✉email: lisa.gunaydin@ucsf.edu

Obsessive-compulsive disorder (OCD) is characterized by recurring, intrusive thoughts (obsessions), and repetitive behaviors (compulsions)[1,2]. Available pharmacological treatments (serotonin reuptake inhibitors or SRIs) rarely produce complete symptom remission and take 2–3 months to produce meaningful symptom relief[3]. Often high SRI doses are required with increased side effect burden, and ~30–40% of OCD patients remain refractory to these treatments[4,5]. Identifying effective, fast-acting treatments will help reduce OCD morbidity.

Increasing evidence indicates that disruptions in glutamate signaling may play a role in OCD symptoms[6,7]. Pioneering human studies have shown that low-dose ketamine, which acts on brain glutamate receptor pathways (among others), has rapid, robust therapeutic effects in depression and other disorders[8–11]. A randomized controlled clinical trial comparing a single low-dose intravenous ketamine to placebo in OCD patients showed ketamine's therapeutic effect was rapid (within hours), and half those who received the drug reported remarkable OCD symptom relief (lasting up to 7 days)[12]. A recent rodent study showed that ketamine rapidly reversed a pharmacologically induced perseverative hyperlocomotion behavior[13]. However, the mechanisms underlying ketamine's therapeutic effect in OCD (and other disorders) are unknown. This knowledge gap is largely due to the fact that no preclinical studies have addressed the circuit effects of ketamine on compulsive behavior in rodent models, a necessary step towards identifying mechanism. Understanding how ketamine modulates neural circuits involved in compulsive behavior will help identify pathways for the rational development of fast-acting therapeutic interventions.

Neuroimaging studies in OCD patients consistently show altered activity of fronto-striatal circuits implicated in action selection, emotion regulation, and cognitive flexibility, although there is conflicting evidence about the directionality of these alterations[14–21]. OCD is associated with dysfunction in two frontal cortical regions: the orbitofrontal cortex (OFC) and dorsal anterior cingulate cortex (dACC). Previous mechanistic work has largely focused on the OFC, a region implicated in assigning value to action outcomes[22]. In contrast, little mechanistic work has addressed the causal role of the dACC in compulsive behavior. The dACC is important for controlling action selection and has also been implicated in fear expression, making it uniquely situated to control both the motor and affective components of OCD[23,24]. The dACC has been associated with conflict monitoring in OCD[25–27], and several structural imaging studies have shown decreased ACC volumes in OCD[28,29]. The fronto-striatal circuit consisting of dACC projections to the dorsal medial striatum (DMS) is of particular interest given its role in controlling cognitive flexibility and goal-oriented behavior, functions that are disrupted in OCD[14,15,30]. Alterations in this circuit may lead to perseverative behaviors by disrupting the balance between goal-oriented and habitual behavior[31–33]. Indeed, several studies support a correlation between increased reliance on habitual learning and severity of compulsions[30,34]. Determining the causal role of the dACC-DMS circuit in compulsive behavior—and how this circuit is modulated by ketamine—could provide critical insights into OCD pathology that are necessary for developing better treatment options.

Rodent studies are beginning to provide insights into corticostriatal mechanisms of OCD-related behavior[35–37]. One rodent model of compulsive behavior is the SAP90/PSD95-associated protein 3 (SAPAP3) knock-out (KO) mouse, which has a prominent compulsive grooming phenotype that is reduced by SRI treatment[38]. Mice lacking SAPAP3 also demonstrate impaired cognitive flexibility, aberrant habit formation, and altered glutamatergic signaling in the dorsal striatum, a region implicated in OCD pathology[38–40]. A recent analysis of post-mortem brains demonstrated reduced expression of the SAPAP3 protein in the striatum of OCD patients, and variants of the *Sapap3* gene have been associated with early-onset OCD and trichotillomania, another compulsive disorder[41]. Previous studies using this genetic model to investigate fronto-striatal circuits have focused on the role of the OFC, with little work addressing the rodent dACC homolog—the dorsomedial prefrontal cortex (dmPFC)[42,43]. Given the importance of the dmPFC-DMS circuit for controlling behavioral flexibility, we hypothesized that alterations in the dmPFC-DMS circuit would be associated with compulsive behaviors in this mouse[31]. The SAPAP3 KO mouse thus provides a platform for testing the causal role of the dmPFC-DMS circuit in compulsive behavior and how novel therapeutics such as ketamine modulate compulsive grooming and its underlying neural circuitry.

Here we demonstrate that ketamine rapidly attenuates compulsive grooming behavior in SAPAP3 KO mice (referred to as "KO mice" from here on). Using in vivo fiber photometry recording from dmPFC-DMS projection neurons, we show that KO mice lack normal modulation of dmPFC-DMS circuit activity associated with grooming behavior in WT mice. Furthermore, ketamine selectively increases activity of this circuit in KO but not WT mice during grooming. Optogenetically mimicking this increased activity by stimulating dmPFC-DMS projections was sufficient to rescue the compulsive grooming phenotype in KO mice. Conversely, optogenetic inhibition of dmPFC-DMS projections increased grooming in WT mice, demonstrating bidirectional control of grooming behavior via this fronto-striatal circuit. These studies demonstrate that ketamine increases activity in a fronto-striatal circuit that causally controls compulsive grooming behavior, suggesting a central role for this circuit in mediating ketamine's therapeutic effects in OCD.

## Results

**Compulsive grooming is associated with blunted fronto-striatal circuit dynamics.** First, we wanted to observe natural neural dynamics in the dmPFC-DMS projection underlying healthy versus compulsive grooming in WT and KO littermates, respectively. We first confirmed that KO mice had increased grooming compared to WT littermates (Fig. 1a, unpaired two-tailed *t*-test, $P = 0.0083$, df = 28; WT = 15, KO = 15). To assess circuit activity during behavior, we used fiber photometry to record calcium signals from fronto-striatal projection neurons expressing the calcium indicator GCaMP6m. To selectively express GCaMP6m in dmPFC-DMS projection neurons, we injected a retrograde canine adenovirus carrying Cre recombinase (CAV2-Cre) into the DMS and a Cre-dependent GCaMP6m virus into dmPFC. We then implanted an optical fiber in the dmPFC to record from this DMS-projecting subpopulation (Fig. 1b, see Supplementary Fig. 1 for detailed histology). We measured the amplitude (Fig. 1c, unpaired two-tailed *t* test $P = 0.2709$, df = 711; WT = 130 grooming epochs from 11 animals, KO = 118 grooming epochs from 10 animals) and frequency (Fig. 1d, unpaired two-tailed *t* test $P = 0.0090$, df = 246; WT = 130 grooming epochs from 11 animals, KO = 118 grooming epochs from 10 animals) of calcium transient peaks during grooming epochs and saw a significant increase in peak frequency in the KO mice. We then generated a z-scored peri-event time histogram (PETH) of dmPFC-DMS projection neuron activity aligned to the onset of grooming behavior (see Supplementary Fig. 2 for examples of raw traces). For generating z-scored PETHs we only used grooming epochs that were separated by a minimum of 5 s from the preceding grooming epoch to reduce any overlap in signal produced by temporally close grooming bouts (see the "Methods" section for details). In WT animals, we observed a decrease in the neural

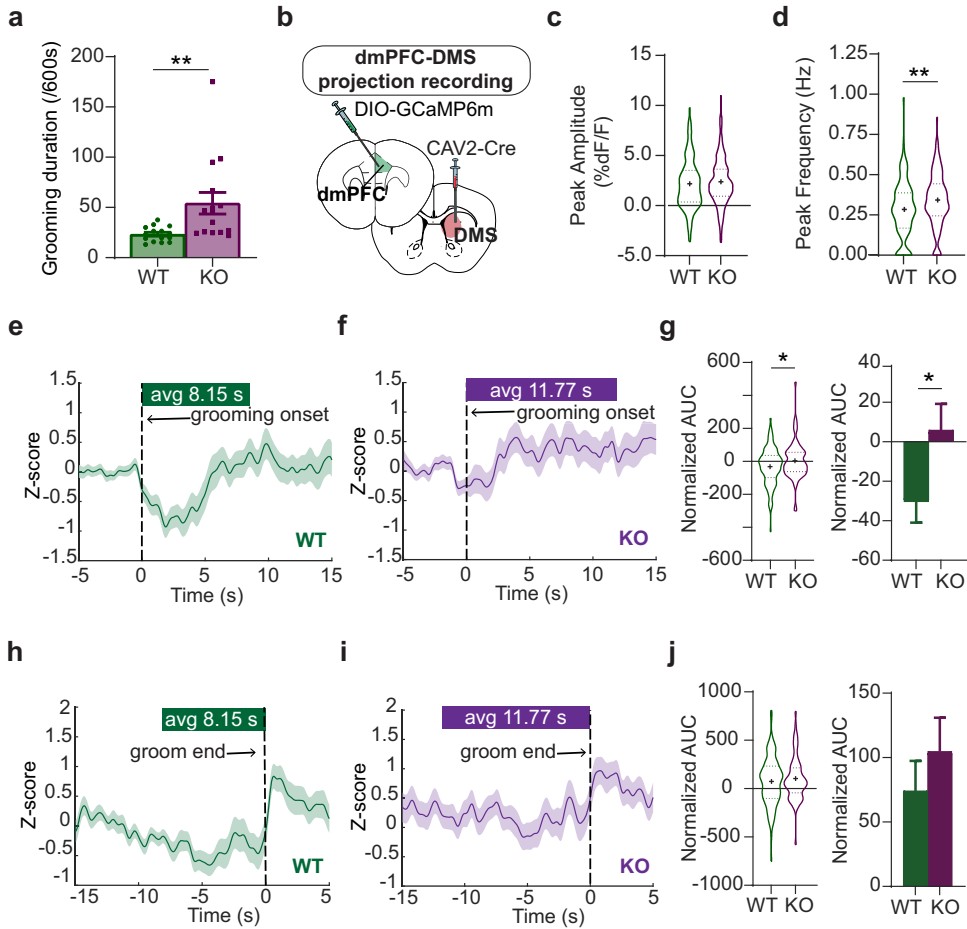

**Fig. 1 Grooming-associated changes in fronto-striatal activity. a** KO mice groom more than WT littermates (unpaired two-tailed *t*-test *P* = 0.0083; WT = 15, KO = 15). **b** Mice are injected with CAV2-Cre in the DMS and Cre-dependent GCaMP6m in the dmPFC, with optical fibers implanted in the dmPFC. **c** Peak amplitude of calcium transients (unpaired two-tailed *t* test, *P* = 0.2709, WT = 130 grooming epochs from 11 mice, KO = 118 grooming epochs from 10 mince). **d** Frequency of calcium transients during grooming (unpaired two-tailed *t*-test P = 0.0090; WT = 130 grooming epochs from 11 animals, KO = 118 grooming epochs from 10 animals). **e** PETH of averaged *z*-score calcium signals aligned to the start of grooming in WT mice (green line, *N* = 113 groom epochs from 11 WTs, green bar above represents average length of WT grooming epochs). **f** KO PETH of averaged *z*-score calcium signals during grooming (purple line, purple bar above represents average length of grooming epochs for KO, *N* = 73 groom epochs from 10 KOs). **g** Normalized area under the curve (AUC) analysis of calcium signal during grooming shown as truncated violin plot (left) and bar graph (right) (unpaired two-tailed *t*-test *P* = 0.0342; WT = 113 groom epochs from 11 animals, KO = 73 groom epochs from 10 animals). **h** PETH of averaged *z*-score signals aligned to the end of grooming in WT mice (*N* = 113 groom epochs from 11 WTs). **i** KO PETH of averaged *z*-score signals aligned to the end of grooming (*N* = 73 groom epochs from 10 KOs). **j** AUC analysis of calcium signal post-grooming shown as truncated violin plot (left) and bar graph (right) (unpaired two-tailed *t*-test *P* = 0.3961, WT = 113 groom epochs from 11 WTs, KO = 73 groom epochs from 10 KOs). Bar graphs and peri-event time histograms (PETH) show data means +/− SEM. Truncated violin plots end at distribution minimum and maximum, dotted lines mark 25% and 75% quartiles, cross denotes mean. Source data are provided as a Source data file. *P* < 0.05, **P* < 0.01.

signal with the start of grooming (Fig. 1e, 113 groom epochs from 11 WT animals). This dip in activity upon grooming onset was absent in KO mice (Fig. 1f, 73 groom epochs from 10 KO animals). To quantify this genotype difference in neural activity during grooming, we analyzed neural activity for the entire grooming duration by analyzing the normalized area under the curve (AUC; calculated for each grooming epoch divided by the duration of that grooming epoch). We saw that WT animals had a significantly more negative normalized AUC compared to KO animals (Fig. 1g, unpaired two-tailed *t* test *P* = 0.0342, df = 184; WT = 113 groom epochs from 11 animals, KO = 73 groom epochs from 10 animals). To assess whether this circuit is also modulated with cessation of grooming, we then focused on the transition out of grooming behavior. We generated PETHs aligned to the end of grooming epochs and saw that both WT (Fig. 1h, 113 groom epochs from 11 WT animals) and KO (Fig. 1i, 73 groom

epochs from 10 KO animals) animals showed an increase in neural activity immediately after the termination of grooming. We quantified this change by calculating the normalized AUC for the 2-s window immediately after the end of a grooming epoch and saw that, unlike during grooming, both WT and KO animals showed an increase in neural activity immediately after grooming ended (Fig. 1j, unpaired two-tailed *t* test *P* = 0.3961, df = 184; WT = 113 groom epochs from 11 animals, KO = 73 groom epochs from 10 animals). In summary, WT animals have a decrease in fronto-striatal activity during grooming. KO animals, however, selectively lack this robust dip in neural activity at grooming onset, suggesting altered encoding of grooming initiation and/or maintenance in this circuit. Furthermore, both WT and KO animals had similar increases in neural activity after grooming ends, potentially indicating that different circuits control grooming initiation and termination in the dmPFC. These

results together support the hypothesis that dmPFC-DMS circuit modulation is involved in normal grooming behavior, and that engagement of this circuit is markedly altered in compulsive groomers.

**Ketamine rescues compulsive grooming behavior.** After establishing baseline differences in dmPFC-DMS circuit activity between normal and compulsive grooming, we tested whether ketamine could rescue the behavioral and circuit deficits observed in SAPAP3 KO mice. We first performed a behavioral pharmacology experiment to determine whether ketamine could attenuate compulsive grooming in KO mice similar to its rapid therapeutic effect observed in human OCD patients. First, we established baseline grooming levels for naive KO mice and their WT littermates. We then divided the mice into two experimental groups, to receive either saline or ketamine (30 mg/kg i.p.) injection. To validate that ketamine injections were successful, we measured locomotor activity for 10 min post-injection (Supplementary Fig. 3a and b), as ketamine has a known acute hyper-locomotor effect within ~20 min of injection[44]. We then recorded grooming behavior and locomotor activity at four additional time points over the course of a week: 1 h, 1 day, 3 days, and 7 days post-injection to assess lasting effects of ketamine on compulsive grooming and locomotion beyond the range of its acute psychomotor effects (Fig. 2a and Supplementary Fig. 3c, respectively). These time points were chosen to mirror the time points at which a therapeutic effect of ketamine was observed in human OCD patients. We quantified both the number of grooming bouts that mice initiated (grooming frequency) as well as the total time spent grooming (grooming duration). At baseline, KO mice had significantly higher grooming frequency than WT littermate controls. Ketamine injection markedly attenuated this compulsive grooming phenotype, restoring normal grooming levels that were indistinguishable from WT animals (Fig. 2b, two-way RM ANOVA: interaction $P = 0.0002$, df = 12, $F = 3.389$; time $P = 0.0250$, df = 4, $F = 2.857$; experimental group $P < 0.0001$, df = 3, $F = 18.54$; subject $P < 0.0001$, df = 46, $F = 3.626$, residual df = 184; see Supplementary Fig. 4 for Tukey's multiple comparisons across experimental groups within a day; WT sal = 14, WT ket = 13, KO sal = 11, KO ket = 12). Specifically, KO mice that received ketamine had significantly reduced grooming frequency at 1 h and 1-day post-injection that was indistinguishable from WT grooming frequency. This attenuated KO grooming gradually returned back to baseline levels by day 3. Ketamine injection had no effect on grooming frequency in WT littermates, and saline injection had no effect on either group. We then plotted these data as a change from baseline to further demonstrate within-genotype effects of ketamine. We saw significant differences between experimental groups and a significant interaction between group and time post-injection. Post hoc analysis revealed that KO-saline and KO-ketamine groups significantly differed in grooming frequency at 1 h and 1 day time points (Fig. 2c, two-way RM ANOVA: interaction $P = 0.0015$, df = 4, $F = 4.819$; time $P = 0.6934$, df = 4, $F = 0.5584$; experimental group $P = 0.0115$, df = 1, $F = 7.661$; subject $P < 0.0001$, df = 21, $F = 3.790$, residual df = 84; KO sal = 11, KO ket = 12), whereas there was no difference in grooming frequency between WT-saline and WT-ketamine groups at any timepoint (Fig. 2d, two-way RM ANOVA: interaction $P = 0.7892$, df = 4, $F = 0.4265$; time $P < 0.0001$, df = 4, $F = 11.25$; experimental group $P = 0.7173$, df = 1, $F = 0.7173$; subject $P < 0.0001$, df = 25, $F = 3.792$, residual df = 100; WT sal = 14, WT ket = 13). A similar pattern of results emerged when we quantified total grooming duration (Fig. 2e, two-way RM ANOVA: interaction $P = 0.0081$, df = 12, $F = 2.344$; time $P = 0.5614$, df = 4, $F = 0.7467$; experimental group $P < 0.0001$, df = 3, $F = 11.05$; subject $P < 0.0001$, df = 46, $F = 5.649$, residual df = 184; see Supplementary Fig. 4 for Tukey's multiple comparisons across experimental groups within a day; WT sal = 14, WT ket = 13, KO sal = 11, KO ket = 12), though grooming duration returned back to baseline levels faster than grooming frequency. A two-tailed paired $t$-test revealed a trend for increase in grooming duration 24 h post-saline injection compared to baseline grooming levels ($P = 0.0814$, df = 10, mean increase in grooming of 43.62 s, 95% confidence interval −6.549 to 93.79). Similar to the grooming frequency data, post hoc analysis of grooming duration revealed that KO-saline and KO-ketamine groups significantly differed in grooming duration at 1 h and 1 day time points (Fig. 2f, two-way RM ANOVA: interaction $P = 0.0208$, df = 4, $F = 3.065$; time $P = 0.7935$, df = 4, $F = 0.4204$; experimental group $P = 0.0040$, df = 1, $F = 10.44$; subject $P = 0.0864$, df = 21, $F = 1.539$, residual df = 84; KO sal = 11, KO ket = 12), whereas there was no difference in WT grooming duration between treatment groups (Fig. 2g, two-way RM ANOVA: interaction $P = 0.4333$, df = 4, $F = 0.9593$; time $P = 0.0069$, df = 4, $F = 3.756$; experimental group $P = 0.7587$, df = 1, $F = 0.09645$; subject $P < 0.0001$, df = 25, $F = 6.101$, residual df = 100; WT sal = 14, WT ket = 13). These data indicate that ketamine selectively attenuates compulsive over-grooming in KO animals without disrupting normal WT grooming behavior.

**Ketamine selectively increases in vivo fronto-striatal projection activity in compulsive groomers.** Previous studies suggest that ketamine increases glutamatergic transmission and alters excitatory/inhibitory balance within the mPFC[45–47]. However, our baseline photometry recordings (Fig. 1) indicated that KO animals had increased fronto-striatal activity compared to WT animals during grooming, leading us to hypothesize that the therapeutic effect of ketamine might be associated with decreasing activity in this projection to resemble WT activity patterns during grooming. We therefore sought to determine how ketamine specifically affects recruitment of fronto-striatal projection neurons during compulsive behavior. We used in vivo fiber photometry to record from dmPFC-DMS projection neurons during grooming, focusing on measuring circuit activity associated with the therapeutic effect of ketamine 24 h post-injection (Fig. 3a). To probe these mechanisms, we first injected mice with saline and then 24 h later analyzed circuit activity during grooming. One week after saline, we injected mice with 30 mg/kg ketamine. This injection order allowed us to assess within-animal circuit changes without concern of potential long-term carryover effects from ketamine exposure that could occur with a Latin squares crossover design, as observed in human studies[12]. We first confirmed the change in grooming induced by ketamine relative to saline injection and saw that KO animals significantly reduced their grooming compared to WT animals post-ketamine (Fig. 3b, unpaired two-tailed $t$-test, $P < 0.0001$, df = 29; WT = 16, KO = 15). We then quantified the peak frequency and peak amplitude of calcium transients during grooming behavior after saline and ketamine injection. We saw no differences between genotypes or drug conditions in the peak frequency during grooming behavior (Fig. 3c, two-way ANOVA: interaction $P = 0.6817$, df = 1, $F = 0.1687$; treatment $P = 0.2201$, df = 1, $F = 1.512$; genotype $P = 0.8585$, df = 1, $F = 0.03184$, residual df = 226; WT = 66 saline groom epochs and 70 ketamine groom epochs from 11 animals, KO saline = 69 groom epochs from 10 animals, KO ketamine = 25 groom epochs from 9 animals). However, in KO animals ketamine significantly increased peak amplitude of calcium transients compared to KO-saline, WT-saline, and WT-ketamine groups during grooming epochs, while

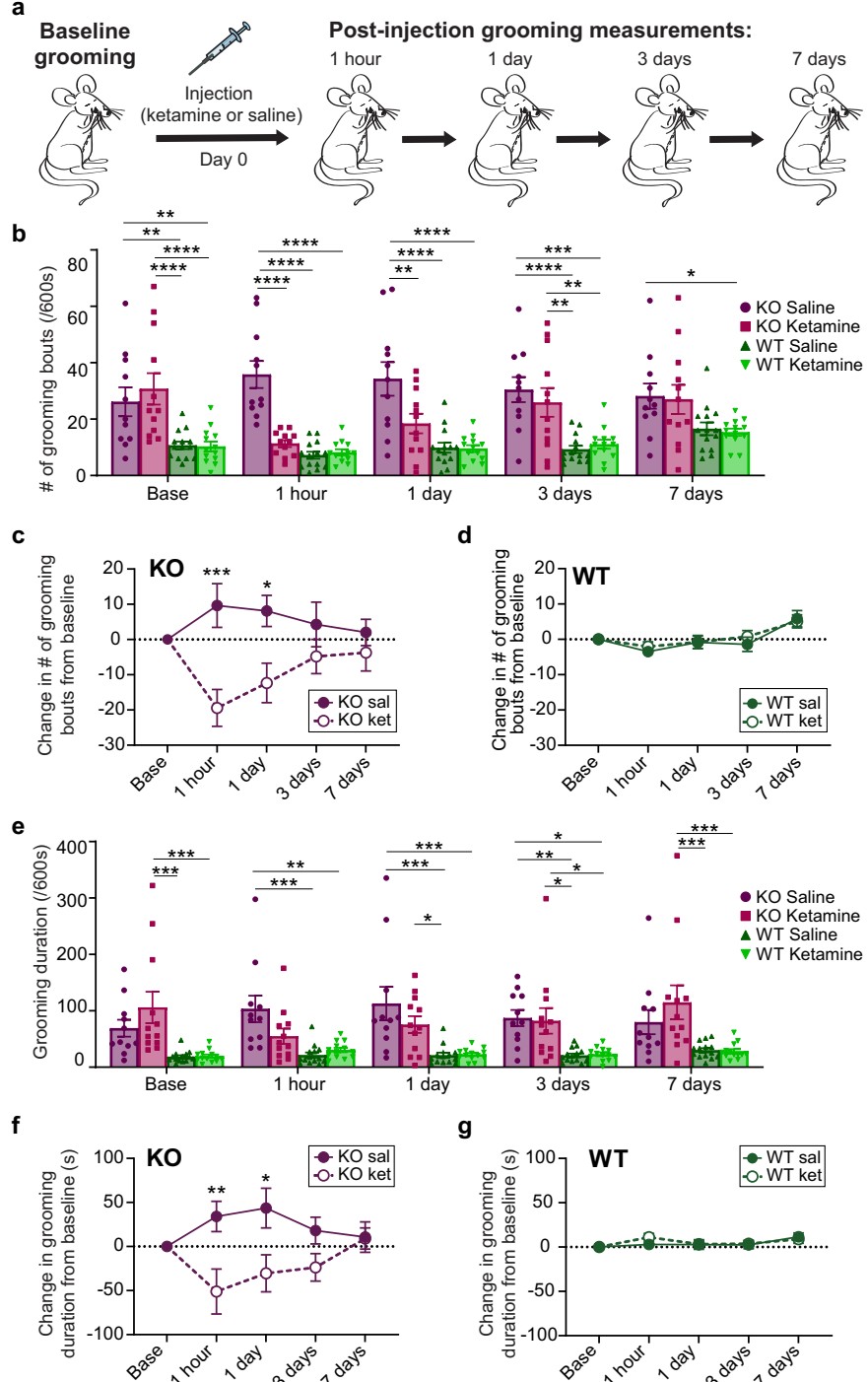

**Fig. 2 Ketamine reduces compulsive grooming in SAPAP3 KO mice. a** Schematic of experiment. **b** KO mice have higher baseline grooming frequency compared to WT controls. Ketamine reduces grooming frequency in KO animals 1-h and 1-day post-injection (two-way RM ANOVA: interaction $P = 0.0002$, time $P = 0.0250$, experimental group $P < 0.0001$, subject $P < 0.001$; significance derived from Tukey's multiple comparisons are marked with asterisk(s); see Supplementary Fig. 4 for exact $P$ values). **c** KO mice given ketamine have reduced grooming frequency compared to KO mice given saline (two-way RM ANOVA: interaction $P = 0.0015$, time $P > 0.05$, experimental group $P = 0.0115$, subject $P < 0.0001$; Sidak's multiple comparisons: 1-h timepoint $P = 0.0002$, 1-day timepoint $P = 0.0138$). **d** WT mice show no change in grooming frequency between treatment groups (two-way RM ANOVA: interaction $P = 0.7892$, time $P < 0.0001$, experimental group $P = 0.7173$, subject $P < 0.0001$). **e** Ketamine attenuates grooming duration in KO animals 1-h post-injection. Saline injection increases KO grooming compared to WT controls 1 h, 1 day, and 3 days post-injection (two-way RM ANOVA: interaction $P = 0.0081$, time $P = 0.5614$, experimental group $P < 0.0001$, subject $P < 0.0001$; significance derived from Tukey's multiple comparisons are marked with asterisk(s); see Supplementary Fig. 4 for exact $P$ values). **f** KO-ketamine mice have decreased grooming duration compared to the KO saline group (two-way RM ANOVA: interaction $P = 0.0208$, time $P = 0.7935$, experimental group $P = 0.0040$, subject $P = 0.0864$; Sidak's multiple comparisons: 1-h timepoint $P = 0.0026$, 1-day timepoint $P = 0.0120$). **g** WT mice displayed no difference in grooming between treatment groups (two-way RM ANOVA: interaction $P = 0.4333$, time $P = 0.0069$, experimental group $P = 0.7587$, subject $P < 0.0001$). Bar and line graphs show data means $+/-$ SEM. WT saline = 14, WT ketamine = 13, KO saline = 11, KO ketamine = 12. Source data are provided as a Source data file. $*P < 0.05$, $**P < 0.01$, $***P < 0.001$, $****P < 0.0001$.

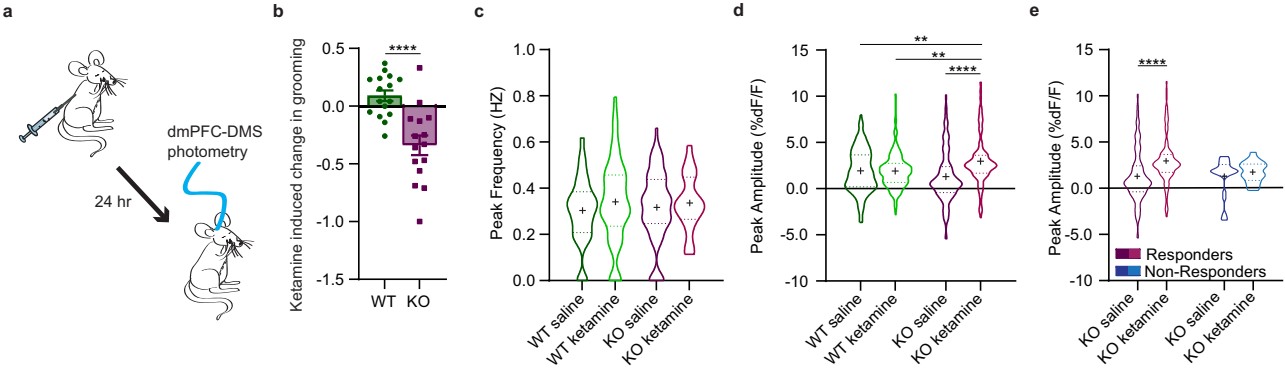

**Fig. 3 Effect of ketamine on fronto-striatal circuit activity. a** Schematic showing fiber photometry recording of dmPFC-DMS projection neurons 24 h after mice are injected with saline or ketamine. **b** KO mice show a significant decrease in grooming duration compared to WT mice after ketamine injection (unpaired two-tailed *t* test *P* < 0.0001, WT = 16 animals, KO = 15 animals). **c** No change in the frequency of calcium transients between saline and ketamine groups (two-way ANOVA; interaction *P* > 0.05, genotype *P* > 0.05, drug *P* > 0.05; WT = 66 saline groom epochs and 70 ketamine groom epochs from 11 animals, KO saline = 69 groom epochs from 10 animals, KO ketamine = 25 groom epochs from 9 animals). **d** Ketamine significantly increases the amplitude of calcium transients during grooming epochs in KO animals but not WT animals 24 h post-injection (two-way ANOVA: interaction *P* < 0.0001, drug *P* < 0.0001, genotype *P* > 0.05; Tukey's multiple comparisons: WT saline vs KO ketamine *P* = 0.0047, WT ketamine vs KO ketamine *P* = 0.0020, KO saline vs KO ketamine *P* < 0.0001; WT = 66 saline groom epochs and 70 ketamine groom epochs from 11 animals, KO saline = 69 groom epochs from 10 animals, KO ketamine = 25 groom epochs from 9 animals). **e** Ketamine does not significantly increase amplitude of calcium transients during grooming epochs in KO mice that do not have a reduction in grooming behavior post-ketamine (two-way ANOVA; interaction *P* > 0.05, treatment *P* = 0.0160, responder-type *P* > 0.05; Tukey's multiple comparisons: responders, KO saline vs KO ketamine *P* < 0.0001; responders, KO saline = 69 groom epochs from 10 animals, KO ketamine = 25 groom epochs from 9 animals; non-responders, KO saline = 6 groom epochs from 2 animals, KO ketamine = 10 groom epochs from 2 animals). Bar graphs and PETHs show data means +/− SEM. Truncated violin plots end at distribution minimum and maximum, dotted lines mark 25% and 75% quartiles, cross denotes data means. Source data are provided as a Source data file. **P < 0.01, ****P < 0.0001.

no change was observed for WT animals between treatment conditions (Fig. 3d, two-way ANOVA: interaction *P* < 0.0001, df = 1, *F* = 19.05; treatment *P* < 0.0001, df = 1, *F* = 17.43; genotype *P* = 0.2720, df = 1, *F* = 1.209, residual df = 673; WT = 66 saline groom epochs and 70 ketamine groom epochs from 11 animals, KO saline = 69 groom epochs from 10 animals, KO ketamine = 25 groom epochs from 9 animals). From our fiber photometry data, we removed 2 KO animals that did not behaviorally respond to ketamine (see "Methods" for exclusion criteria). KO mice that did not respond to ketamine treatment did not show the same ketamine-induced increase in peak amplitude compared to the SAPAP3 KO mice that did respond to ketamine, though this difference could be due to low sample number of non-responders (Fig. 3e, two-way ANOVA: interaction *P* = 0.1699, df = 1, *F* = 1.891; treatment *P* = 0.0160, df = 1, *F* = 5.852; responder-type *P* = 0.1748, df = 1, *F* = 1.848, residual df = 390; responders, KO saline = 69 groom epochs from 10 animals, KO ketamine = 25 groom epochs from 9 animals; non-responders, KO saline = 6 groom epochs from 2 animals, KO ketamine = 10 groom epochs from 2 animals). Next, we generated PETHs of dmPFC-DMS projection activity aligned to grooming onset. WT-saline and WT-ketamine groups both showed a dip in neural activity at grooming onset (Fig. 4a, 51 groom epochs under saline and 55 groom epochs under ketamine from 11 WT animals). The KO-saline group appeared to have a subtle dip in neural activity at grooming onset, whereas the KO-ketamine group showed a large increase in neural activity at grooming onset (Fig. 4b, 46 groom epochs under saline from 10 KO animals and 17 groom epochs under ketamine from 9 KO animals). Quantifying the normalized AUC (as in Fig. 1g) confirmed that the KO-ketamine group had significantly increased neural activity during grooming compared to KO-saline, WT-saline, and WT-ketamine groups (Fig. 4c, two-way ANOVA: interaction *P* = 0.0592, df = 1, *F* = 3.609; treatment *P* = 0.0110, df = 1, *F* = 6.609; genotype *P* = 0.0129, df = 1, *F* = 6.318, residual df = 165; WT = 51 saline groom epochs and 55 ketamine groom

epochs from 11 animals, KO saline 46 groom epochs from 10 animals, KO ketamine = 17 groom epochs from 9 animals). We then generated PETHs aligned to the end of grooming epochs and saw that WT-saline and WT-ketamine groups had a similar increase in neural activity immediately after the termination of grooming behavior (Fig. 4d). KO-saline mice lacked this post-grooming increase in activity, while KO-ketamine mice maintained their increased neural activity observed during grooming (Fig. 4e). To quantify these differences, we calculated the normalized AUC for the 2-s window immediately after the end of grooming. We saw a significant increase in the AUC upon grooming end for SAPAP3 KO-ketamine mice compared to SAPAP3 KO-saline mice (Fig. 4f, two-way ANOVA: interaction *P* = 0.0243, df = 1, *F* = 5.171; treatment *P* = 0.0096, df = 1, *F* = 6.869; genotype *P* = 0.6671, df = 1, *F* = 0.6671, residual df = 165; WT = 51 saline groom epochs and 55 ketamine groom epochs from 11 animals, KO saline 46 groom epochs from 10 animals, KO ketamine = 17 groom epochs from 9 animals). These results indicate that loss of SAPAP3 sensitizes dmPFC-DMS circuitry to ketamine-induced increases in activity, while this circuitry appears resistant to such effects in WT animals. In addition, because ketamine did not restore KO dmPFC-DMS projection activity to WT levels, but instead further increased activity in this projection, it is possible that the higher fronto-striatal activity we observed during pre-injection baseline grooming (Fig. 1d and g) may represent a compensatory mechanism in KO animals.

Next, we performed ex vivo slice electrophysiology recordings from dmPFC neurons 24 h after ketamine or saline injection to determine if ketamine altered their intrinsic excitability. These recordings showed no effect of genotype or drug treatment on the intrinsic excitability of dmPFC neurons (Supplementary Fig. 5), supporting the notion that ketamine exerts its behavioral effect via altered in vivo recruitment of dmPFC neurons during grooming. To determine whether ketamine injection instead alters the release properties or synaptic strength in dmPFC-DMS

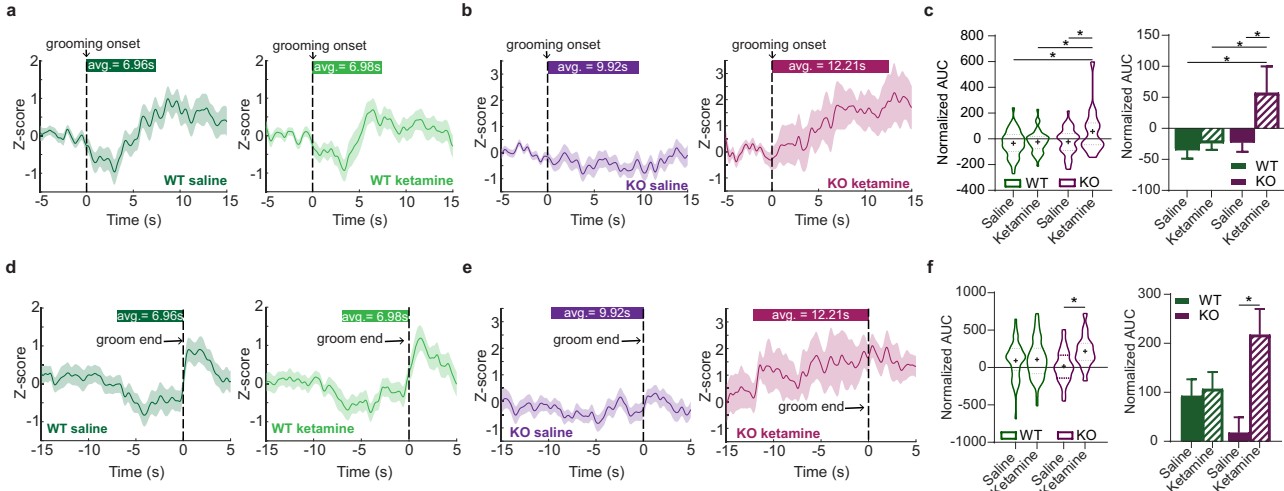

**Fig. 4 Additional effects of ketamine on fronto-striatal circuit activity. a** PETHs of averaged z-score signals aligned to grooming onset (green bar represents average length of grooming epoch) after saline (left, $N = 51$ groom epochs from 11 WTs) and ketamine injections (right, $N = 55$ groom epochs from 11 WTs) in WT animals. **b** PETHs of averaged z-score signals aligned to grooming onset in KO animals after saline (left, $N = 46$ groom epochs from 10 KOs) and ketamine injections (right, $N = 17$ groom epochs from 9 KOs). **c** Area under the curve (AUC) quantification for each grooming epoch shown as truncated violin plot (left) and bar graph (right). Two-way ANOVA; interaction $P = 0.0592$, genotype $P = 0.0129$, drug effect $P = 0.0110$; Tukey's multiple comparison test: WT saline vs KO ketamine $P = 0.0113$, WT ketamine vs KO ketamine $P = 0.0313$, KO saline vs KO ketamine $P = 0.0407$; WT = 51 grooming epochs with saline and 55 grooming epochs with ketamine from 11 animals, KO saline = 46 grooming epochs from 10 animals, KO ketamine = 17 grooming epochs from 9 animals). **d** PETHs of averaged z-score signals aligned to the end of grooming in WT mice after saline (left, $N = 51$ groom epochs from 11 WTs) and ketamine (right, $N = 55$ groom epochs from 11 WTs). **e** PETHs of averaged z-score signals aligned to the end of grooming in KO mice after saline (left, $N = 46$ groom epochs from 10 KOs) and ketamine (right, $N = 17$ groom epochs from 9 KOs). **f** AUC analysis of neural signal after groom end shown as truncated violin plot (left) and bar graph (right). Two-way ANOVA: interaction $P = 0.0243$, genotype $P > 0.05$, treatment $P = 0.0096$; Tukey's multiple comparisons: KO saline vs KO ketamine $P = 0.0180$). Bar graphs and Peri-event time histograms (PETHs) show means $+/-$ SEM. Truncated violin plots end at distribution minimum and maximum, dotted lines mark 25%–75% quartiles, cross denotes means. Source data are provided as a Source data file. *$P < 0.05$.

projections, we injected channelrhodopsin-2 (ChR2) under control of the CaMKIIα promoter into the dmPFC followed by ex vivo slice recordings in the DMS performed 24 h after saline or ketamine injection. First, we assessed release probability by optogenetically stimulating the dmPFC projections and measuring the paired-pulse ratio (PPR). Interestingly, ketamine increased PPR in slices from WT animals, despite no observable behavioral or fiber photometry differences between WT-saline and WT-ketamine mice. In addition, slices from KO animals regardless of treatment showed elevated PPR compared to WT-saline, potentially indicating reduced release probability in dmPFC-DMS neurons of KO mice and WT-ketamine mice (Supplementary Fig. 6a–c). To assess synaptic strength at dmPFC-DMS synapses, we also performed recordings of asynchronous excitatory post-synaptic currents (aEPSCs) in strontium after a single pulse of light stimulation. KO-saline animals had lower aEPSC frequency than WT-saline animals, with no difference in aEPSC amplitude. Ketamine reduced this genotype difference in aEPSC frequency, indicating that ketamine increases aEPSC frequency in KO mice (Supplementary Fig. 6d–h). These data suggest that increased release frequency at dmPFC-DMS synapses may contribute to the increased activity observed in this projection in our in vivo fiber photometry recordings.

**Optogenetic inhibition of dmPFC-DMS projections induces over-grooming in WT animals.** Our observation that WT animals have a dip in dmPFC-DMS projection activity at the onset of grooming suggested that reduced activity in this circuit may be permissive of grooming. To test this hypothesis, we used optogenetic inhibition to decrease activity in the dmPFC-DMS projection and examine whether this manipulation could causally

increase grooming behavior in WT animals. We injected WT mice in the dmPFC with either an eYFP control virus or halorhodopsin (eNpHR3.0) under control of the CaMKIIα promoter and implanted optical fibers in the DMS to bilaterally inhibit fronto-striatal terminals (Fig. 5a, see Supplementary Fig. 7 for detailed histology). Grooming behavior was scored during a 20-min session consisting of a 5-min baseline period, followed by 10 min with the laser on, and an additional 5 min after the laser was turned off. When the laser was turned on to inhibit the dmPFC-DMS projection, grooming behavior increased in the eNpHR3.0 group, whereas eYFP mice maintained constant levels of grooming across the entire experimental time window (Fig. 5b, 8 mice per group). The eNpHR3.0 group showed a shorter latency to their first grooming bout after the laser was turned on (Fig. 5c, unpaired two-tailed t-test, $P = 0.0150$, df = 14; 8 mice per group), but still had a delay of ~30 s to grooming onset. The fact that this increase in grooming behavior was not time-locked to the onset of the laser suggests that inhibiting this projection is permissive of grooming rather than directly driving the behavior. We averaged the amount of time spent grooming across 5-min epochs of laser on versus laser off and saw that in the eNpHR3.0 group grooming significantly increased for the first 5 min while the laser was on (Fig. 5d, two-way RM ANOVA: interaction $P = 0.0395$, df = 3, $F = 3.036$; laser $P = 0.0841$, df = 3, $F = 2.370$; virus $P = 0.0010$, df = 1, $F = 17.31$; subject $P = 0.1264$, df = 14, $F = 1.577$, residual df = 42; 8 mice per group). Analysis revealed a significant main effect of virus with a significant interaction between virus and laser. At baseline, there were no grooming differences between eYFP and eNpHR3.0 groups. Mice expressing eNpHR3.0 but not eYFP had a significant laser-evoked increase in grooming duration from baseline (Fig. 5e, unpaired two-tailed t-test, $P = 0.0083$, df = 14; 8 mice per group).

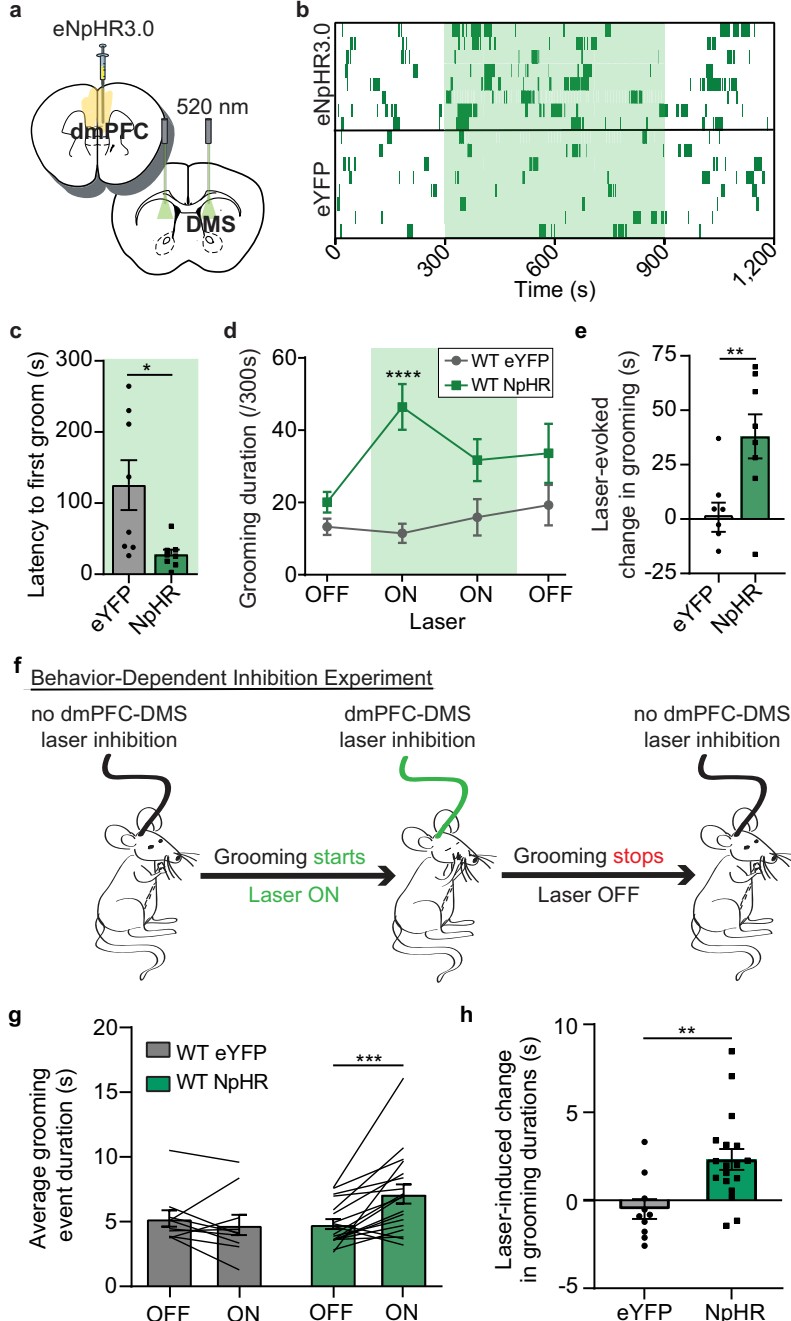

**Fig. 5 Optogenetic inhibition of dmPFC-DMS projections increases grooming in WT mice. a** WT mice are bilaterally injected with CaMKIIα-eNpHR3.0 virus in the dmPFC and implanted bilaterally with optical fibers in the DMS. **b** Raster-like plot of grooming behavior for individual animals across experimental timeline (top: eNpHR3.0; bottom: eYFP control). Each row represents one animal. Dark green dashes indicate grooming timestamps and light green shading indicates when laser was on (8 mice per group). **c** eNpHR3.0 mice began grooming significantly sooner than eYFP mice after laser onset (unpaired two-tailed $t$-test $P = 0.0150$). **d** eNpHR3.0 mice ($N = 8$) show increased grooming during laser on epochs that is significantly higher than eYFP controls ($N = 8$) (two-way RM ANOVA: interaction $P = 0.0395$, laser $P = 0.0841$, virus group $P = 0.0010$, subject $P = 0.1264$; Sidak's multiple comparisons: eNpHR3.0 vs eYFP first laser on epoch $P < 0.0001$). **e** Bar graph quantifying data from (**d**) showing that eNpHR3.0 mice have a significant laser-evoked increase in grooming compared to eYFP mice (unpaired two-tailed $t$ test $P = 0.0083$). **f** graphical representation of behavior-dependent inhibition experiment. **g** Mice expressing eNpHR3.0 ($N = 18$) show an average increase in the duration of grooming epochs when the laser is turned on during grooming that is not seen in eYFP-WT mice ($N = 10$) (two-way RM ANOVA: interaction $P = 0.0043$, laser $P = 0.0526$, virus $P > 0.5$, subject $P = 0.0010$; Sidak's multiple comparisons: eNpHR3.0 Off vs eNpHR3.0 On $P = 0.0004$). Black lines represent individual animals. Lines start and stop at individual animal's average grooming event duration during laser off and on conditions. **h** Laser inhibition of dmPFC-DMS projections significantly extends the average duration of grooming epochs in NpHR-WT mice ($N = 18$) compared to eYFP-WT mice ($N = 10$) (unpaired two-tailed $t$-test $P = 0.0043$). Bar and line graphs show data means $+/-$ SEM. Source data are provided as a Source data file. *$P < 0.05$, **$P < 0.01$, ***$P < 0.001$, ****$P < 0.0001$.

Next, we took advantage of the temporal precision of optogenetics to perform a "behavior-dependent" inhibition experiment in which we investigated whether we could extend individual grooming events in WT mice by briefly inhibiting the dmPFC-DMS projection when animals start grooming. Grooming behavior was scored during a 20-min session consisting of a 5 min baseline with no laser, followed by 5 min during which the start of each grooming event would trigger the laser to stay on until the mouse stopped grooming (Fig. 5f). This was followed by another 5 min laser-off period, and then a final 5 min period of our behavior-dependent inhibition protocol. We then generated the average duration of grooming events for laser-off versus behavior-dependent laser-on epochs. Mice expressing eNpHR3.0 but not eYFP had a significant increase in the average duration of grooming events when the laser was briefly turned on during grooming (Fig. 5g, two-way RM ANOVA: interaction $P = 0.0043$, df = 1, $F = 9.779$; laser $P = 0.0526$, df = 1, $F = 4.124$; virus $P = 0.2587$, df = 1, $F = 1.333$; subject $P = 0.0010$, df = 26, $F = 3.536$, residual df = 26; eYFP = 10 mice, NpHR = 18 mice). We then calculated the laser-induced change in grooming duration and saw that eNpHR3.0 mice had a significant increase in the duration of grooming epochs compared to eYFP mice during behavior-dependent inhibition (Fig. 5h, unpaired two-tailed *t*-test, $P = 0.0043$, df = 26; eYFP = 10 mice, NpHR = 18 mice). These results indicate that inhibiting the dmPFC-DMS projection produces a hyper-grooming phenotype and that acute inhibition during an individual grooming event can prolong grooming in WT mice.

**Stimulation of dmPFC-DMS projections rescues compulsive grooming in SAPAP3 KO animals.** Given that the therapeutic effect of ketamine was associated with increased neural activity in fronto-striatal projection neurons during grooming in KO animals, we reasoned that optogenetically elevating activity in this circuit might likewise have a therapeutic effect on compulsive grooming. We injected channelrhodopsin-2 (ChR2) under control of the CaMKIIα promoter or the eYFP control virus (AAV5-CaMKIIa-eYFP) into the dmPFC and implanted an optical fiber in the DMS (Fig. 6a, see Supplementary Fig. 8 for detailed histology). First, we performed a behavior-dependent stimulation protocol where grooming behavior was scored during a 20 min session consisting of four 5 min blocks that alternated between laser off and laser stimulation that was contingent upon grooming (i.e., turned on at the start of grooming and turned off when the mouse stopped grooming) (Fig. 6b). We then calculated the average duration of grooming events for laser was off versus laser on epochs. KO mice expressing ChR2 but not eYFP had a significant decrease in the average duration of grooming events when the laser was briefly turned on during grooming (Fig. 6c, two-way RM ANOVA: interaction $P = 0.0252$, df = 1, $F = 6.695$; laser $P = 0.0241$, df = 1, $F = 6.830$; virus $P = 0.4258$, df = 1, $F = 0.6840$; subject $P = 0.0043$, df = 11, $F = 5512$, residual df = 11; eYFP = 6 KO mice, ChR2 = 7 KO mice). We then calculated the laser-induced change in grooming duration and saw that ChR2-KO mice had a significant decrease in the duration of grooming events compared to eYFP-KO mice (Fig. 6d, unpaired two-tailed *t*-test, $P = 0.0252$, df = 11; eYFP = 6 KO mice, ChR2 = 7 KO mice). These results together demonstrate that using brief closed loop stimulation and inhibition of the dmPFC-DMS projection bidirectionally decreases and increases, respectively, the duration of individual grooming events.

Next, we performed a standard time-dependent stimulation protocol to determine the effects of continuous stimulation on overall grooming behavior. We recorded grooming behavior for a 5-min baseline period, a 5-min laser on period (10 Hz stimulation), and a final 5-min laser off period (Fig. 6e). At baseline, we saw a significant difference in grooming levels between WT-ChR2 and KO-ChR2 animals. When the laser was on, KO-ChR2 grooming behavior was reduced to WT levels. Upon termination of the laser, KO-ChR2 grooming returned to significantly higher levels than the WT-ChR2 mice, demonstrating that stimulation of dmPFC terminals in the DMS had an acute therapeutic effect on aberrant KO grooming behavior (Fig. 6f, two-way RM ANOVA: interaction $P = 0.0430$, df = 2, $F = 3.311$; laser $P < 0.0001$, df = 2, $F = 22.22$; genotype $P = 0.0018$, df = 1, $F = 11.64$; subject $P < 0.0001$, df = 31, $F = 4.936$, residual df = 62; WT-ChR2 = 18, SAPAP3 KO-ChR2 = 15). Interestingly, stimulation of dmPFC terminals did not alter the number of grooming bouts that KO mice engaged in, demonstrating that stimulation of dmPFC only altered duration of grooming (Supplementary Fig. 9). Laser stimulation had no effect on grooming of KO-eYFP controls or WT-ChR2 and WT-eYFP groups (Fig. 6g, one-way ANOVA; group $P = 0.0104$, df = 3, $F = 4.163$, residual df = 50; WT-eYFP = 11, WT-ChR2 = 18, SAPAP3 KO-eYFP = 10, SAPAP3 KO-ChR2 = 15). Since stimulating this projection had an acute effect of attenuating compulsive grooming, we tested whether repeated stimulation of this projection could produce a lasting therapeutic effect. Mice received three additional stimulation sessions over 2 weeks, totaling 50 min of laser stimulation (see "Methods" for details). We then scored grooming behavior in the absence of laser 4 days after the last stimulation to look for lasting therapeutic effects. KO-ChR2 cumulative grooming behavior was indeed significantly reduced after this repeated stimulation compared to pre-stimulation baseline grooming behavior, an effect that was not present in KO-eYFP, WT-eYFP, and WT-ChR2 groups (Fig. 6h, two-way RM ANOVA: interaction $P = 0.0354$, df = 3, $F = 3.149$; group $P < 0.0001$, df = 3, $F = 10.04$; time $P = 0.0070$, df = 1, $F = 8.088$; subject $P = 0.0190$, df = 40, $F = 1.946$, residual df = 40; WT-eYFP = 11, WT-ChR2 = 13, SAPAP3 KO-eYFP = 10, SAPAP3 KO-ChR2 = 10). Specifically, when we compared the change in grooming duration after repeated stimulation between KO-ChR2 and WT-ChR2 groups, we saw that KO-ChR2 had a significantly greater reduction in grooming behavior relative to WT-ChR2 (Fig. 6i, unpaired two-tailed *t*-test, $P = 0.0230$, df = 21; WT-ChR2 = 13, SAPAP3 KO-ChR2 = 10). These optogenetic experiments together demonstrate causal control of compulsive grooming behavior via manipulation of the dmPFC-DMS projection. Furthermore, these results indicate that minimal repeated stimulation of this projection may have lasting therapeutic effects on compulsive behavior.

**Ketamine blocks exacerbation of SAPAP3 KO grooming phenotype induced by optogenetic inhibition.** Our observation that optogenetically stimulating dmPFC-DMS projections rescues the compulsive grooming behavior in KO mice supported our hypothesis that ketamine may exert its therapeutic effect on grooming behavior through increased dmPFC-DMS activity. In addition, our baseline fiber photometry data showed that KO mice, prior to ketamine, have increased dmPFC-DMS activity which we hypothesized may represent an incomplete compensatory mechanism to try to limit grooming behavior. We therefore further reasoned that (1) inhibiting the dmPFC-DMS projection would exacerbate grooming behavior in KO mice, and (2) that ketamine should block this increase in grooming. To provide more direct evidence in support of this hypothesis, we next sought to combine optogenetic inhibition with ketamine administration to test whether ketamine would occlude the optogenetic effect. We therefore optogenetically inhibited

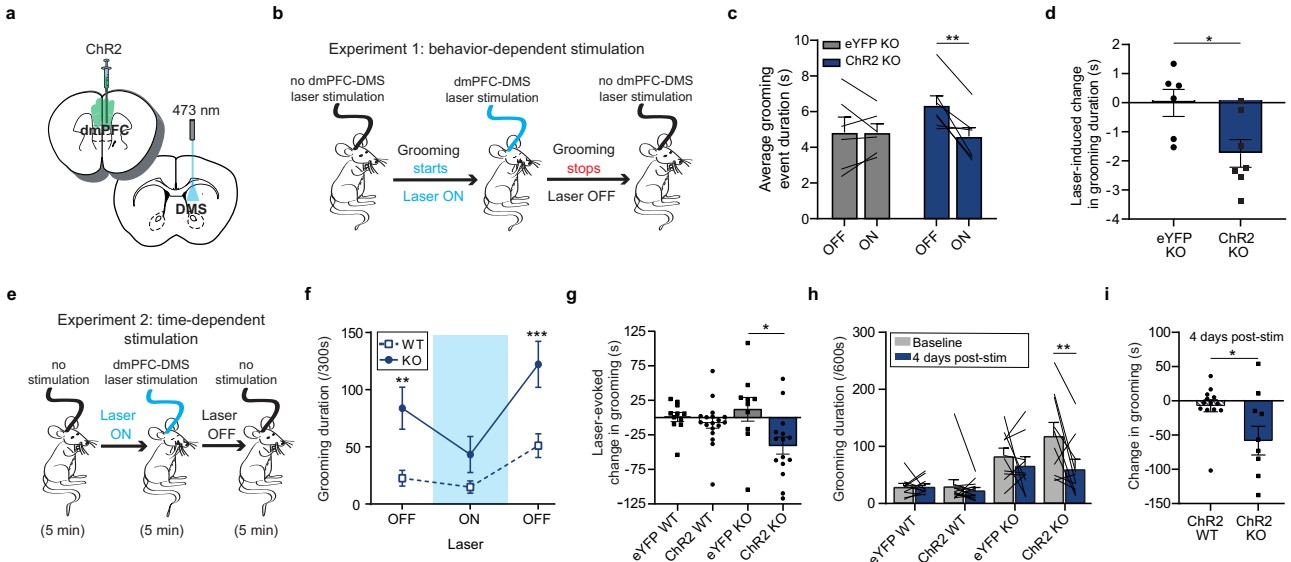

**Fig. 6 Optogenetic stimulation of dmPFC-DMS projections reduces compulsive grooming in SAPAP3 KO mice.** Bar and line graphs show data means +/− SEM. Source data are provided as a Source data file. *$P < 0.05$, **$P < 0.01$, ***$P < 0.001$. Black lines on bar graph's represent individual animals. Lines start and stop at individual animal's average for that graph's conditions. **a** Cartoon of viral injection and implant. **b** Graphical representation of behavior-dependent stimulation experiment (ChR2-KO mice = 7, eYFP-KO mice = 6). **c** Duration of grooming epochs during behavior-dependent stimulation (two-way RM ANOVA: interaction $P = 0.0252$, laser $P = 0.0241$, virus $P > 0.05$, subject $P = 0.0043$; Sidak's multiple comparisons: ChR2-KO on vs off $P = 0.0056$). **d** Laser-evoked change in average duration of grooming epochs (unpaired two-tailed t-test $P = 0.0252$). **e** Graphical representation of time-dependent stimulation experiment (ChR2-KO = 15, ChR2-WT = 18, eYFP-KO = 10, eYFP-WT = 11). **f** Grooming duration across 5-min blocks (two-way RM ANOVA: interaction $P = 0.0430$, laser $P < 0.0001$, genotype $P = 0.0018$, subject $P < 0.0001$; Sidak's multiple comparisons: baseline ChR2-WT vs ChR2-KO $P = 0.0043$, laser off ChR2-WT vs ChR2-KO $P = 0.0007$). **g** Laser-evoked reduction in grooming behavior relative to baseline (one-way ANOVA $P = 0.0104$; Tukey's multiple comparisons: eYFP-KO vs ChR2-KO $P = 0.0120$). **h** Following 2 weeks of repeated stimulation, KO-ChR2 mice have a significant and sustained reduction in grooming behavior 4 days after the last laser stimulation session (two-way RM ANOVA: interaction $P = 0.0354$, time $P = 0.0070$, experimental group $P < 0.0001$, subject $P = 0.0190$; Sidak's multiple comparisons: ChR2-KO baseline vs 4 days post-stimulation $P = 0.0015$). **i** Four days after the final stimulation, KO-ChR2 mice show a significantly greater decrease from baseline grooming behavior than WT ChR2 mice (unpaired two-tailed t test $P = 0.0230$).

dmPFC-DMS activity in KO mice 24 h after they received either saline or ketamine (Fig. 7a, see Supplementary Fig. 7c for detailed histology). We hypothesized that inhibiting dmPFC-DMS activity would increase grooming behavior in KO mice, while ketamine would counteract this effect. First, we scored baseline grooming behavior in eYFP-KO and NpHR-KO mice. We then injected mice with either saline or ketamine and made behavioral recordings 1 h post-injection to ensure that ketamine had its expected behavioral effect on reducing grooming behavior. Both eYFP-KO and NpHR-KO mice injected with ketamine showed a significant reduction in grooming duration compared to baseline grooming levels (Fig. 7b, two-way RM ANOVA: interaction $P = 0.0181$, df = 3, $F = 4.230$; treatment $P = 0.1692$, df = 3, $F = 1.858$; time $P = 0.0003$, df = 1, $F = 18.68$; subject $P = 0.0027$, df = 20, residual df = 20; saline eYFP-KO = 5, ketamine eYFP-KO = 6, saline KO-NpHR = 6, ketamine KO-NpHR = 7). KO-NpHR mice also showed a significant reduction in grooming frequency post-ketamine (Fig. 7c, two-way RM ANOVA: interaction $P = 0.0278$, df = 3, $F = 3.738$; treatment $P = 0.5286$, df = 3, $F = 0.7619$; time $P = 0.0045$, df = 1, $F = 10.23$; subject $P = 0.0260$, df = 20, $F = 2.445$, residual df = 20; saline KO-eYFP = 5, ketamine KO-eYFP = 6, saline KO-NpHR = 6, ketamine KO-NpHR = 7). Twenty-four hours post-injection we recorded grooming behavior for 15 min, including a 5-min baseline period, a 5-min laser on period (continuous light), and a final 5-min laser off period. During the laser on period NpHR-KO mice that received saline showed significantly higher grooming behavior compared to NpHR-KO mice that received ketamine and eYFP-KO mice (Fig. 7d, two-way RM ANOVA: interaction $P = 0.0221$, df = 6, $F = 2.819$; laser

$P = 0.0167$, df = 2, $F = 4.537$; treatment $P = 0.0612$, df = 3, $F = 2.886$; subject $P = 0.0037$, df = 20, $F = 2.698$, residual df = 40; saline KO-eYFP = 5, ketamine KO-eYFP = 6, saline KO-NpHR = 6, ketamine KO-NpHR = 7). We observed no significant alterations in grooming frequency (Fig. 7e, two-way RM ANOVA: interaction $P = 0.6185$, df = 6, $F = 0.7428$; laser $P = 0.1310$, df = 2, $F = 2.139$; treatment $P = 0.2568$, df = 3, $F = 1.455$; subject $P = 0.0004$, df = 20, $F = 3.472$, residual df = 40; saline KO-eYFP = 5, ketamine KO-eYFP = 6, saline KO-NpHR = 6, ketamine KO-NpHR = 7). This experiment demonstrated that the KO grooming phenotype can be exacerbated by inhibiting PL-DMS activity, lending support to our hypothesis that the increased activity observed in the KOs baseline fiber photometry recordings is a compensatory mechanism. Furthermore, ketamine's ability to block the increase in grooming caused by inhibiting PL-DMS activity provides additional support to the hypothesis that the ketamine-induced increase in PL-DMS activity reduces grooming behavior in KO mice.

## Discussion

Cortico-striatal circuits are heavily implicated in the pathophysiology of OCD, although few previous mechanistic studies in rodents have interrogated the role of the dmPFC-DMS circuit in compulsive behavior, despite its known role in OCD-relevant processes such as cognitive flexibility and action-outcome learning. Here we identify how activity of the dmPFC-DMS circuit is altered in SAPAP3 KO mice that display compulsive grooming behavior. Furthermore, we describe how ketamine affects compulsive grooming and its underlying dmPFC-DMS circuit

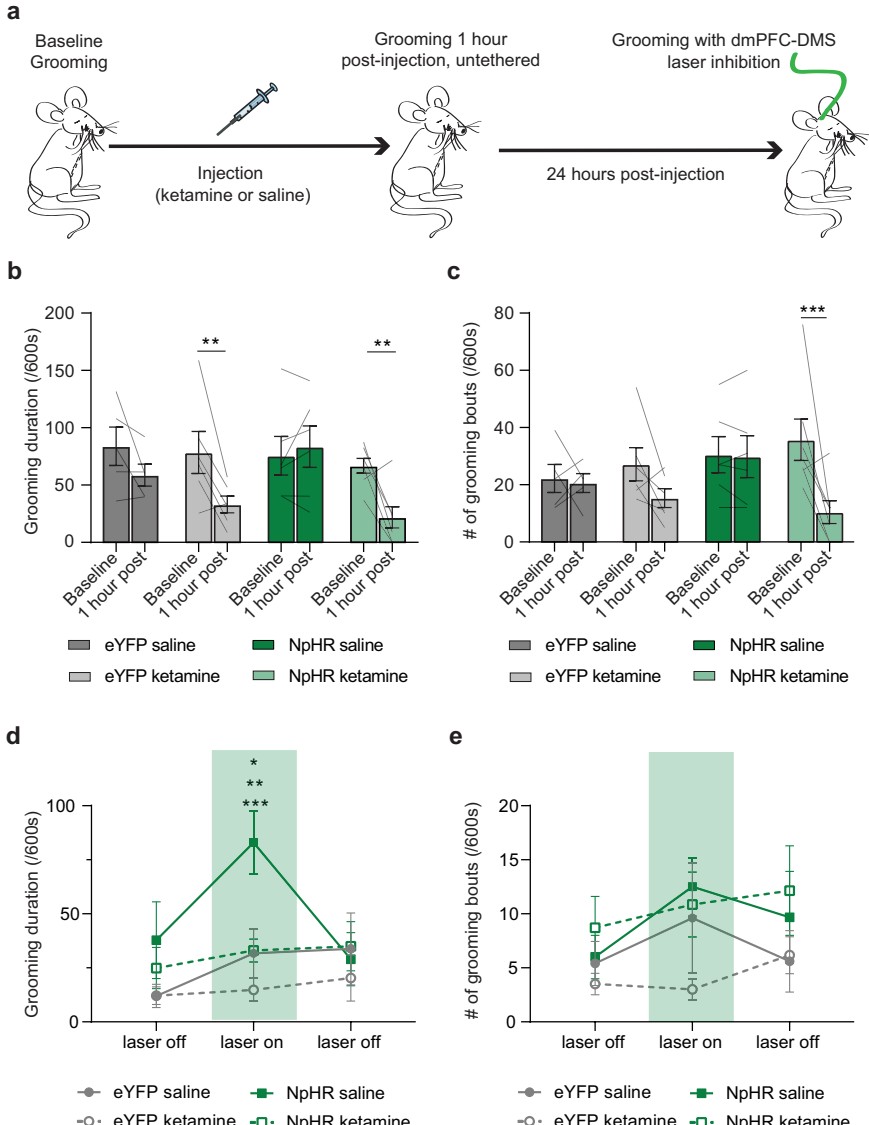

**Fig. 7 Ketamine blocks grooming increase caused by inhibition of dmPFC-DMS projections in SAPAP3 KO mice.** Bar and line graphs show data means +/− SEM. Black lines on bar graphs represent individual animals. Lines start and stop at individual animal's average for that graph's conditions. Source data are provided as a Source data file. *$P < 0.05$, **$P < 0.01$, ***$P < 0.001$. **a** SAPAP3 KO with bilateral implants in the DMS expressing either eYFP or eNpHR3.0 in the dmPFC are injected with either saline or ketamine followed by grooming measurements. **b** Ketamine reduces grooming duration 1-h post-injection (two-way RM ANOVA: interaction $P = 0.0181$, treatment $P > 0.05$, time $P = 0.0003$, subject $P = 0.0027$; Sidak's multiple comparisons: ketamine KO-eYFP baseline vs 1-h $P = 0.0061$, ketamine KO-NpHR baseline vs 1-h $P = 0.0033$; saline KO-eYFP = 5, ketamine KO-eYFP = 6, saline KO-NpHR = 6, ketamine KO-NpHR = 7). **c** Only SAPAP3 KO-NpHR that receive ketamine show a significant reduction in grooming frequency 1-h post-injection (two-way RM ANOVA: interaction $P = 0.0278$, treatment $P > 0.05$, time $P = 0.0045$, subject $P = 0.0260$; Sidak's multiple comparisons: ketamine KO-NpHR baseline vs 1-h $P = 0.0009$; saline KO-eYFP = 5, ketamine KO-eYFP = 6, saline KO-NpHR = 6, ketamine KO-NpHR = 7). **d** Inhibition of dmPFC-DMS projections significantly increases grooming in SAPAP3 KO-NpHR mice injected with saline (two-way RM ANOVA: interaction $P = 0.0221$, laser $P = 0.0167$, treatment $P = 0.0612$, subject $P = 0.0037$; Tukey's multiple comparisons: saline KO-eYFP vs saline KO-NpHR $P = 0.0147$, ketamine KO-eYFP vs saline KO-NpHR $P = 0.0003$, ketamine KO-NpHR vs saline KO-NpHR $P = 0.0086$; saline KO-eYFP = 5, ketamine KO-eYFP = 6, saline KO-NpHR = 6, ketamine KO-NpHR = 7). **e** Neither inhibition of dmPFC-DMS projections nor drug treatment affectgrooming frequency (two-way RM ANOVA: interaction $P > 0.05$, laser $P > 0.05$, treatment $P > 0.05$, subject $P = 0.0004$; saline KO-eYFP = 5, ketamine KO-eYFP = 6, saline KO-NpHR = 6, ketamine KO-NpHR = 7).

modulation. In this study, we use ketamine as a tool to probe fronto-striatal circuit mechanisms underlying compulsive grooming, and demonstrate ketamine's therapeutic effect in rodent models of compulsive behavior (another study recently demonstrated ketamine reversal of a pharmacologically induced perseverative hyperlocomotion)[13]. We show that ketamine provides fast-acting therapeutic relief in SAPAP3 KO mice, reducing both the frequency of grooming bouts and cumulative time spent

grooming. This therapeutic effect remained significant 24 h after ketamine injection and then tapered off by day 3 post-injection. The time course of this anti-compulsive effect of ketamine in SAPAP3 KO mice was strikingly similar to that observed in clinical studies of ketamine in OCD, which found immediate symptom relief that lasted up to 7 days[12]. Our rodent results thus corroborate human studies identifying ketamine as a promising candidate for a fast-acting pharmacological treatment for OCD.

These experiments establish the SAPAP3 KO mice as a preclinical translational model for testing ketamine metabolites and related pharmacologic agents to screen for drugs that may have a similar therapeutic profile for OCD-like behavior with fewer side effects[48,49].

To determine circuit changes associated with the anti-compulsive effect of ketamine, we used in vivo fiber photometry recording of fronto-striatal projection neurons to assess genotype- and drug-related differences in dmPFC-DMS circuit activity during grooming. Our baseline fiber photometry recordings showed several key differences in fronto-striatal circuit activity between WT and SAPAP3 KO mice. WT fronto-striatal projection neurons had a drop-in activity at the onset of grooming, but no such modulation was observed in SAPAP3 KO mice. The sustained neural activity during grooming observed in SAPAP3 KO mice appeared to be driven by an increased frequency of calcium transients compared to WT mice. Interestingly, upon transition out of grooming, both WT and SAPAP3 KO mice showed an increase in neural activity relative to pre-grooming baseline levels. This could indicate that, with respect to grooming behavior, dmPFC-DMS projection neurons have two states (low activity permits grooming whereas high activity suppresses grooming). Loss of the low activity state with maintenance of the high activity state in SAPAP3 KO mice could represent compensatory mechanisms in the dmPFC to try to modulate compulsive grooming behavior. Furthermore, these two activity states could be driven by different synaptic inputs onto the dmPFC, which may be differentially dysregulated in SAPAP3 KO mice. In contrast to these in vivo circuit differences, ex vivo prefrontal slice recordings did not reveal any genotype differences in basal cellular properties of dmPFC neurons. Other studies have demonstrated altered endocannabinoid signaling in the striatum of SAPAP3 KO mice, which could affect local release properties of glutamatergic projections without changing overall prefrontal cell body activity[50]. Optical stimulation of dmPFC terminals during our ex vivo DMS slice recordings indicated that ketamine administration did not alter presynaptic release probability, as measured by paired-pulse ratios (PPR), in SAPAP3 KO mice. Ketamine did, however, increase PPR in WT mice which was unexpected given the lack of both behavioral and fiber photometry effects of ketamine in WT mice. Ketamine also mitigated the differences between KO and WT mice in the frequency of asynchronous release from dmPFC synaptic terminals, which could indicate increased dmPFC synaptic activity in SAPAP3 KO-ketamine mice. This would be in line with our fiber photometry results, where ketamine increased in vivo dmPFC-DMS activity. Interestingly, recent work has demonstrated that protein kinase A activation can increase the number of vesicles released by regulating the readily releasable pool without altering release probability, showing increased frequency in aEPSCs without altering PPR[51]. Of note, we do not have post-injection behavioral data for the animals used in our slice experiments and so the variability in this data set could potentially be accounted for by KO-ketamine non-responders. Given our fiber photometry data showing that SAPAP3 KO-ketamine non-responders do not have an increase in dmPFC-DMS activity, the potential inclusion of such non-responders in our slice physiology experiments could occlude more robust ketamine-induced changes in our slice data. Overall, these results suggest that ketamine likely exerts its behavioral effect on grooming not via altered intrinsic excitability of dmPFC cells, but rather via altered in vivo recruitment of dmPFC neurons during compulsive behavior. Furthermore, our slice results support the hypothesis that the in vivo activity differences we observed between WT and SAPAP3 KO mice could result in increased multivesicular release or increased release frequency of dmPFC-DMS projections in the SAPAP3 KO mice. However, we cannot exclude the possibility

that ketamine administration may also alter presynaptic inputs onto dmPFC-DMS neurons, contributing to the ketamine-induced increase in dmPFC-DMS activity. For example, in another mouse model of compulsivity, repetitive checking behavior is driven by reduced mPFC glutamatergic activity due to feed-forward inhibition from BLA inputs[52].

Based on our baseline neural recordings, we originally hypothesized that ketamine administration in SAPAP3 KO animals might restore the dip in fronto-striatal activity observed in WT animals during grooming. Instead, ketamine significantly increased dmPFC-DMS projection activity in SAPAP3 KO but not WT animals. This suggests that the lack of a grooming-related dip in neural activity during baseline recordings in SAPAP3 KO mice may represent compensatory activity in this projection to counteract hyperactive downstream motor circuitry (e.g., DLS or central striatum) that directly drives compulsive grooming[53], while the ketamine-induced increase in glutamatergic transmission may allow for dmPFC-DMS projection neurons to more successfully restore normal behavior. Furthermore, it seems unlikely that the WT dip in fronto-striatal activity is directly driving grooming behavior but may instead represent a permissive signal for grooming. The timing of our optogenetic inhibition effect corroborates this idea: inhibiting the dmPFC-DMS projection in WT mice did not immediately drive grooming behavior when the laser was turned on, but rather gradually increased the probability of grooming episodes, consistent with a permissive role of this circuit. In addition, saline injection, which appeared to exacerbate grooming duration in SAPAP3 KO animals 24 h post-injection, reduced SAPAP3 KO fronto-striatal activity during grooming, further supporting a model in which decreased fronto-striatal projection activity promotes the initiation of grooming behavior. Therefore, the dmPFC-DMS projection neurons may represent a permissive signal that normally promotes grooming behavior through decreased activity, but is disrupted in SAPAP3 KO mice. A recent study demonstrated that cortico-striatal limbic loops, such as the dmPFC-DMS projection, are capable of modulating cortico-striatal motor loops, providing a potential anatomical substrate for permissive dmPFC-DMS gating of DLS that can be causally tested in future studies[54]. The additional boost in prefrontal glutamatergic signaling provided by ketamine may therefore further dampen the dmPFC's permissive signal to a threshold that is sufficient to overcome dysregulated downstream grooming motor circuitry and restore healthy grooming levels[55]. Indeed, both the pharmacological (ketamine) and optogenetic manipulations we employed that increase activity in the dmPFC-DMS projection had a convergent effect of decreasing compulsive grooming in SAPAP3 KO mice (Fig. 8a), while optogenetic inhibition of the dmPFC-DMS projection increased grooming in both WT (Fig. 8b) and SAPAP3 KO mice. Furthermore, the exacerbation of the SAPAP3-KO grooming phenotype caused by optogenetic inhibition of the dmPFC-DMS terminals supports our hypothesis that the elevated dmPFC-DMS activity we see in our baseline data is a compensatory mechanism. Importantly, administration of ketamine prevented the increase in grooming behavior in the SAPAP3 KO mice induced by inhibiting the dmPFC-DMS terminals (Fig. 8c), further supporting a role of increasing dmPFC-DMS circuit activity in ketamine's therapeutic effect on compulsive grooming behavior. Interestingly, others have demonstrated that reductions in neural activity of dmPFC projections to the brainstem could induce compulsive alcohol drinking behavior in mice, supporting the idea that disengagement of the dmPFC contributes to compulsive action[56].

Several rodent studies have highlighted the importance of the orbitofrontal cortex, ventral striatum, and amygdalo-striatal projection in compulsive behavior[35,52]. We now add to this growing body of scientific work by demonstrating the causal role

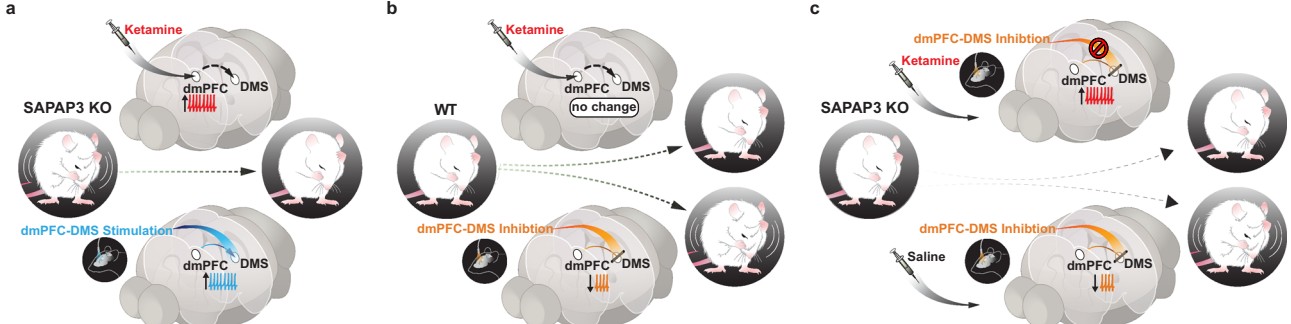

**Fig. 8 Increasing dmPFC-DMS projection neuron activity via ketamine or optogenetic stimulation rescues compulsive grooming behavior.** Graphical summary of the paper showing: **a** Ketamine produces a decrease in grooming behavior that is correlated with an increase in dmPFC-DMS projection neuron activity in SAPAP3 KO mice. This rescue of compulsive grooming behavior is causally reproduced by selectively stimulating dmPFC-DMS projection neurons. **b** Ketamine has no effect on grooming behavior or dmPFC-DMS circuit activity in WT mice, though optogenetic inhibition of dmPFC-DMS projections in WT mice was sufficient to induce increased grooming. **c** Ketamine blocks the increase in grooming behavior induced by optogenetic inhibition of dmPFC-DMS projection in KO mice.

of the dmPFC-DMS circuit in compulsive behavior. In our study, we demonstrate that bidirectional manipulation dmPFC-DMS activity can acutely alter grooming behavior in both WT and SAPAP3 KO mice. Our rescue of compulsive grooming in the SAPAP3 KO mice by increasing dmPFC-DMS activity mirrors the acute therapeutic effect on compulsive behavior seen in SAPAP3 KO mice when lateral orbitofrontal (OFC) terminals are stimulated in the dorsal striatum[36]. It is possible that increasing dmPFC and lateral OFC projection activity in the striatum both act to offset the increased striatal drive that SAPAP3 KO mice receive from the secondary motor cortex[53]. Of particular interest, however, is that our repeated stimulation of dmPFC-DMS projections induced a long-lasting reduction in grooming 4 days after the final stimulation session. Others have demonstrated similar lasting changes in compulsive behavior through chronic manipulations of fronto-striatal circuitry. Of note, repeated stimulation of projections to the ventromedial striatum from ventral and medial OFC in WT mice produce a delayed but sustained increase in grooming behavior, mimicking a compulsive grooming phenotype[35]. Our finding together with this previous work suggests that stimulating fronto-striatal projections may induce robust and lasting plasticity at these synapses. Further studies into the mechanisms of such fronto-striatal plasticity may be of particular importance for understanding the etiology of OCD and for designing novel therapeutic interventions. By dissecting how each of these circuits contributes to compulsive behavior, we can begin to build a more complete picture and better identify points of convergence for developing targeted therapeutic intervention. Furthermore, we provide the first steps in understanding how ketamine can interact with circuits altered in OCD to restore normal goal-directed behavior, providing a preclinical platform for testing neural circuit mechanisms underlying novel fast-acting treatments for OCD.

## Methods
**Animals**. All experiments were performed under a protocol approved by the Institutional Animal Care and Use Committees at the University of California, San Francisco. Homozygous SAPAP3 knockout (KO) mice and their wild-type (WT) littermates were generated from breeding two heterozygotes that were maintained on a C57BL/6J background. For a subset of optogenetic experiments, wild-type C57BL/6J mice purchased from The Jackson Laboratory were also used. The over-grooming phenotype in SAPAP3 KOs emerges around 4 months of age, so all mice were at least 120 days old (average age = 290 days). Both male and female mice were used (see Supplementary Fig. 10 for sex distributions and age information for each experimental cohort). In addition, because of the grooming variability that occurs in SAPAP3 KO mice average grooming levels were compared across behavioral cohorts. No significant differences were seen in grooming levels across cohorts (Supplementary Fig. 11). All mice were raised in

normal light conditions (12:12 light/dark cycle), with ad libitum access to food and water. Mouse colony rooms were kept at an ambient temperature of 69–73 °F with 50–65% humidity. Experiments were performed during the inactive cycle.

**Surgery**. Mice were anesthetized using 5% isoflurane at an oxygen flow rate of 1 L/min and placed in a stereotaxic apparatus (Kopf Instruments) on top of a heating pad. Anesthesia was maintained with 1.5–2.0% isoflurane for the duration of surgery. Respiration and toe pinch response were monitored closely. Slow-release buprenorphine (0.5 mg/kg) and ketoprofen (1.6 mg/kg) were administered subcutaneously at the start of surgery. The incision area was shaved and cleaned with ethanol and betadine. Lidocaine (0.5%) was administered topically. An incision was made along the midline, and bregma and lambda were measured to level the skull. Virus was injected and fiber optic ferrule(s) implanted as described below. After securing the ferrule with dental cement, the skin was sutured around the implant and mice recovered in a clean cage atop a heating pad. The following day, mice were monitored for healthy recovery and administered a subsequent dose of ketoprofen.

*Inhibition of dmPFC-DMS projections*. To optogenetically inhibit projections from dmPFC to DMS, 500 nL of AAV5-CaMKIIα-eNpHR3.0-eYFP or AAV5-CaMKIIα-eYFP control virus (UNC Vector Core) was bilaterally injected into the mPFC (+/−0.35 M/L; 1.8 A/P; −2.6 D/V, in mm from bregma) using a 10 μl nanofil syringe (World Precision Instruments) with a 33 gauge beveled needle at a rate of 0.1 μl/min. After waiting 15 min, the needle was slowly withdrawn. Two optical fibers in 1.25 mm ceramic ferrules (0.39 NA, 200-μm diameter, Thorlabs) were then implanted bilaterally in the DMS (+/−1.2 M/L; 0.8 A/P; −3 D/V). Fibers were lowered slowly into the DMS and ferrules were secured to the skull using dental cement (C&B Metabond). These surgeries were done on C57BL/6J WT mice (The Jackson Laboratory), and on SAPAP3 KO animals and WT littermates. For both inhibition and stimulation experiments of dmPFC-DMS projections AAV5-CaMKIIα-eYFP virus, which lacks the light-responsive opsins, was used as the control for surgical procedures, non-specific effects of AAV5 viral expression, and any non-opsin specific behavioral changes induced by light changes when lasers are switched on and off.

*Stimulation of dmPFC-DMS projections*. For stimulation of projections from dmPFC to DMS, 500 nL of AAV5-CaMKIIα-ChR2-eYFP (Addgene) or AAV5-CaMKIIα-eYFP control virus (UNC Vector Core) (diluted 1:3 in saline) was injected unilaterally into the left dmPFC and an optical fiber was implanted unilaterally in the DMS (balanced across hemispheres), using the same coordinates and ferrules as described above. These surgeries were done on SAPAP3 KO animals and WT littermates.

*Fiber photometry*. For recording from dmPFC neurons that project to the DMS, 700 nL of a 50/50 mixture of AAV1.hSyn.mCherry (to visualize injection location; UNC Vector Core) and CAV2-Cre (Plateforme de Vectorologie de Montpellier) was injected unilaterally into the DMS (+/−1.4 M/L; 0.8 A/P; −3.5 D/V). To allow CAV2-Cre-induced GCaMP expression in the dmPFC, 1500 nL of AAV1.Syn.-Flex.GCaMP6m.WPRE.SV40 (Addgene) was unilaterally injected into the dmPFC (+/−0.35 M/L; 1.9 A/P; −2.5 D/V). An optical fiber stub in a 2.5 mm ceramic ferrule (0.48 NA, 400-μm diameter, Doric Lenses) was then implanted into the dmPFC (+/−0.35 M/L; 1.9 A/P; −2.3 D/V) and secured as described above. These surgeries were done on SAPAP3 KOs and WT littermates. GCaMP6m expression was assessed post-perfusion by counting and then averaging the number of GCaMP6m cells present in two brain slices per animal. Images of the implant location were taken with a 6D Widefield Nikon Ti Inverted Microscope. An area of

400 microns by 400 microns was defined directly under the implant site, and cells expressing GCaMP6m were manually counted. The average number of neurons expressing GCaMP6m in this area was 45.5 neurons +/− 10 s.e.m.

**Viruses**. pAAV.Syn.Flex.GCaMP6m.WPRE.SV40 (viral titer $2.10 \times 10^{13}$ vg/mL) was a gift from The Genetically Encoded Neuronal Indicator and Effector Project (GENIE) & Douglas Kim (Addgene viral prep # 100838-AAV1; RRID:Addgene_100838). pAAV-CaMKIIa-hChR2(H134R)-EYFP (viral titer $2.20 \times 10^{13}$ vg/mL) was a gift from Karl Deisseroth (Addgene viral prep # 26969-AAV5; RRID:Addgene_26969). AAV5-CaMKIIa-eNpHR3.0-eYFP (viral titer $4.9 \times 10^{12}$ vg/mL) was a gift from Karl Deisseroth and packaged by the UNC Vector Core. AAV5-CaMKIIa-eYFP (viral titer $3.60 \times 10^{12}$ vg/mL) was a gift from Karl Deisseroth and packaged by the UNC Vector Core. AAV1-hSyn.mCherry (viral titer $4.10 \times 10^{12}$ vg/mL) was a gift from Karl Deisseroth and packaged by the UNC Vector Core. CAV-2-cre virus (viral titer $1.00 \times 10^{13}$ pp/mL) was from Plateforme de Vectorologie de Montpellier.

**Behavioral assays**

*Grooming*. To observe grooming behavior, mice were placed in a clear plastic cylinder (36-cm diameter) and recorded with two video cameras to have both top and side views. All behavioral assays were done in an open top sound dampened chamber. A blind observer manually scored and timestamped grooming behavior. Locomotor activity (distance traveled, velocity) was recorded and quantified by Ethovision XT software (Noldus). A TTL pulse triggered by the Ethovision XT software was used to synchronize fiber photometry recording data with behavioral data.

*Behavioral pharmacology*. Grooming behavior for SAPAP3 KOs and WTs was recorded for 10-min sessions as described above. Baseline grooming behavior was measured, and then mice were split into two groups to receive either saline (0.9% sterile sodium chloride) or ketamine (30 mg/kg, Sigma). The next day mice were injected intraperitoneally with either saline or ketamine and immediately placed in the behavioral rig. Locomotor recordings were taken for 10 min post-injection to verify successful ketamine injection through induction of a short-term increase in locomotor behavior. No grooming measurements were taken at this time. Grooming was scored at additional behavioral time points of 1 h, 1 day, 3 days, and 7 days post-injection. It should be noted that prior to using the 30 mg/kg dose a small cohort of animals were tested with a 20 mg/kg dose of ketamine (Supplementary Fig. 12). Since no effect was seen at 20 mg/kg the dose was increased to 30 mg/kg which was partly based on dosing schemes used by others to understand ketamine's rapid anti-depressant effects in mice[48].

*Optogenetics*. For all optogenetic experiments, a Master-8 (A.M.P.I.) pulse generator controlled by TTL pulses from Ethovision (Noldus) software was used to drive the laser. Laser output was delivered to the animal via an optical fiber (0.39 NA, 200 μm, Thorlabs) connected to a 1x1 fiber optic rotary joint (Doric Lenses) followed by another optical fiber (0.37 NA, 200 μm, Doric Lenses) which was coupled to the cannula through a ceramic sleeve (Thorlabs). For the time-dependent stimulation protocol, baseline measurements were recorded while mice were not connected to a patch cord and were observed moving freely. To stimulate projections from dmPFC to DMS using ChR2, blue light was generated by a 473 nm laser (Shanghai Laser & Optics Century Co. LTD) (0.5–0.8 mW, 10 Hz, 5 ms pulse width). Each session was divided into three epochs: a 5-min baseline period followed by 5 min of laser on and 5 min of laser off post-stimulation. A subset of animals received 3 additional days of stimulation totaling 50 min spread across 2 weeks. After the time-dependent stimulation experiment mice were rested for 2 days before additional stimulation, then rested for 3 days, and then 2 days. In addition, these extra stimulation sessions occurred in three different environments (open field, an elevated zero maze, mouse cage with marbles). We then assessed the long-term effects of the laser on grooming behavior 4 days after the final stimulation when mice were untethered. This experiment was performed 6 weeks post-surgery. For the behavior-dependent stimulation protocol (0.8 mW, 10 Hz, 5 ms pulse width) each session was divided into four epochs: 5 min no laser period, 5 min period where the laser was turned on and off with the initiation and termination of grooming, respectively, 5 min no laser period, and 5 min period where the laser was turned on and off with the initiation and termination of grooming, respectively. The laser was triggered on and off by TTL pulses sent by the Ethovision XT software (Noldus). The TTL pulses were controlled by manual keyboard strokes at the start and end of grooming. This experiment was performed 6–9 weeks post-surgery.

To inhibit projections from dmPFC to DMS using eNpHR3.0, green light was generated by a 532 nm laser (Shanghai Laser & Optics Century Co. LTD) (5 mW continuous light on each side). For time-dependent bilateral inhibition of dmPFC-DMS terminals in WT animals, each trial began with a 5-min baseline period followed by 10 min of laser on and 5 min of laser off post-stimulation. This experiment was performed ~7 weeks post-surgery. The behavior-dependent bilateral inhibition of dmPFC-DMS terminals in WT animals followed the same protocol as the described above for the behavior-dependent stimulation protocol and occurred 11 weeks post-surgery. For bilateral inhibition of dmPFC-DMS terminals in KO animals, untethered baseline grooming measures were recorded for 10 min. KO mice then received either an injection of saline (10 mg/mL) or ketamine (30 mg/kg) and their grooming behavior was recorded untethered 1 h

post-injection. Twenty-four hours post-injection grooming behavior was recorded across a 5-min baseline period followed by 5 min of laser on and 5 min of laser off post-stimulation. This experiment was performed 11 weeks post-surgery.

*Fiber photometry*. Prior to surgery we quantified grooming behavior over 10 min for each animal to confirm the increased grooming phenotype in the SAPAP3 KO mice. Mice were allowed to recover from surgery for 4–5 weeks before fiber photometry recordings. For dmPFC-DMS fiber photometry recordings, SAPAP3 KOs and WT littermates underwent 2 days of recording baseline grooming behavior. Each recording session was 20 min. Following the end of recording on the second day of baseline, mice received an injection of saline solution (10 mg/mL, i.p.) and were returned to their home cage. Behavioral measurements and fiber photometry recordings then occurred 24 h later. After 7 days post-saline injection mice were then administered a dose of ketamine (30 mg/kg, i.p.) and returned to their home cage. Behavioral measurements and fiber photometry recordings were taken 24 h after ketamine injection. Fiber photometry signals were demodulated and analyzed using custom MATLAB code. Briefly, the 405 nm signal was regressed against the 470 nm signal and the polyfit MATLAB function (least-squares method) was used to determine the coefficients to calculate a fitted 405 nm signal. The DF/F was then calculated ((490 nm signal − fitted 405 nm signal)/(fitted 405 nm signal)). For peak amplitude and frequency calculations, we first detected all $Ca^{2+}$ transient peaks throughout the signal using custom peak detection code using a running average method to calculate the peak to trough value. We used a 10 s trough window (window during convolution for finding running average trough) and a 1 s temporal window (minimum amount of time between peaks). Once peaks were detected, we then calculated the average frequency and amplitude of these peaks. We restricted the analysis of grooming to epochs longer than 3 s. Prior to analysis of the photometry signals post-injection, we determined which KO mice were responders versus non-responders to ketamine by calculating a grooming index to assess change in grooming behavior induced by ketamine ((grooming duration post-ketamine − grooming duration post-saline)/(grooming duration post-ketamine + grooming duration post-saline). We then generated the average and standard deviation for WT animals. KO animals had to have a reduction in grooming behavior post-ketamine was at least 1 standard deviation different from the change in grooming behavior of WT animals to be included in the fiber photometry analysis. Only 2 out of 15 KO mice did not meet this criterion. We analyzed the number of calcium transients and the amplitude of these transients that occurred during grooming. We also generated peri-event time histograms (PETH) of the neural activity aligned to the beginning and end of grooming bouts and generated normalized area under the curve analysis (AUC) for each grooming epoch (for the entirety of the grooming epoch) and for the 2-s window immediately after grooming ended. In order for grooming bouts to be included in the PETH analysis, we implemented a 5 s threshold in which the start of the grooming epoch had to be at least 5 s after the end of the prior grooming epoch. This allowed us a 5-s window prior to groom start that we knew did not have any signal interference from other grooming epochs in that we used to generate the necessary baseline mean signal used to convert the photometry signals into z-score.

In vivo calcium imaging data were acquired using a custom-built rig based on a previously described setup from the Deisseroth lab[57]. This setup was controlled by an RZ5P fiber photometry processor (TDT) and Synapse software (TDT). The RZ5P/Synapse software controlled a 4-channel LED Driver (DC4100, Thorlabs), which in turn controlled two fiber-coupled LEDs: 470 nm for GCaMP stimulation and 405 nm to control for artifactual fluorescence (M470F3, M405FP1, Thorlabs). These LEDs were sinusoidally modulated at 210 Hz (470 nm) and 320 Hz (405 nm) and connected to a Fluorescence Mini Cube with 4 ports (Doric Lenses), and the combined LEF output was connected through a fiber optic patch cord (0.48 NA, 400 μm, Doric Lenses) to the cannula via a ceramic sleeve (Thorlabs). The emitted light was focused onto a Visible Femtowatt Photoreceiver Module (Model 2151, Newport, DC low) and sampled at 60 Hz. Noldus behavioral analysis was synchronized to the photometry setup using TTL pulses generated every 10 s following the start of the Noldus trial.

**Perfusions**. Following the conclusion of our behavioral experiments, animals were anesthetized using 5% isoflurane and given a lethal dose (1.0 ml) cocktail of ketamine/xylazine (10 mg/ml ketamine, 1 mg/ml xylazine). They were then transcardially perfused with 10 mL of 1X PBS followed by 10 mL 4% paraformaldehyde (PFA). Brains were left in 4% PFA overnight and then transferred to a 30% sucrose solution until slicing. The brains were frozen and sliced on a sliding microtome (Leica Biosystems) and placed in cryoprotectant in a well-plate. Slices were then washed in 1X PBS, mounted on slides (Fisherbrand Superfrost Plus), and air dried (covered). ProLong Gold antifade reagent (Invitrogen, ThermoFisher Scientific) was pipetted on top of the slices, a cover slip (Slip-rite, ThermoFisher) was placed on top, and the slides were left to dry overnight (covered). Viral injection, fiber photometry cannula implant, and optogenetic cannula implant placements were histologically verified on a fluorescence microscope (Leitz DMRB, Leica). Neural and behavioral data of mice with incorrectly targeted fiber photometry implants were removed from analysis.

**Statistical analysis**. Statistical analyses were performed using GraphPad Prism 8 software package. Statistical significance was set at $P < 0.05$ for all experimental results. The type of statistical analysis used was determined independently for each experiment and is listed in the relevant results section along with sample size, exact $p$ values, degrees of freedom (df) and $F$ values. Figure captions also provide

statistical information including information on post hoc multiple comparisons tests where appropriate. Bar and line graphs represent data-set means and error bars represent standard error of the mean (s.e.m.).

**Reporting summary**. Further information on research design is available in the Nature Research Reporting Summary linked to this article.

## Data availability

The datasets generated during and analyzed during the current study are available from the corresponding author upon reasonable request. Source data are provided with this paper.

## Code availability

The code used for the fiber photometry analysis is available from the corresponding author on reasonable request.

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

## Acknowledgements

We would like to thank Julia Kuhl for her exceptional graphic design work in generating the graphical summary for Fig. 7, and Teagan Bullock for assistance with brain slicing and targeting on a subset of animals. We thank Dr. Boscardin from the UCSF CTSI Consulting service for biostatistics advice. For histology and targeting figures we used The Mouse Brain atlas as a reference for generation of brain slice images with landmarks (Paxinos G., Franklin K.B.J. (2007). The mouse brain in stereotaxic coordinates. Compact 3rd Ed. Elsevier Academic Press). This study was supported by a Chan-Zuckerburg Biohub Award (Dr. Gunaydin). It was also supported in part by the IOCDF Breakthrough Award (Dr. Rodriguez), Robert Wood Johnson Harold Amos Medical Faculty Development Program (Dr. Rodriguez), R01MH105461 (Dr. Rodriguez), and by the National Center for Advancing Translational Sciences, National Institutes of Health, through UCSF-CTSI (NIH/NCATS UL1 TR001872).

## Author contributions

G.L.D., C.I.R. and L.A.G. all contributed to the intellectual conceptualization and design of the overall study. G.L.D. designed, performed, and analyzed data for the behavioral pharmacology, fiber photometry, brief stimulation optogenetic experiments, and KO ketamine optogenetic experiments and helped analyze the remaining optogenetic experiments. A.R.M. designed, performed, and analyzed optogenetic experiments. A.L. performed and analyzed slice electrophysiology experiments in the dmPFC. L.D.S. performed and analyzed slice electrophysiology experiments in the DMS. G.L.D. and L.A.G. wrote the manuscript and made the figures.

## Competing interests

In the last 3 years, Dr. Rodriguez has served as a consultant for Epiodyne and received research grant support from Biohaven Pharmaceuticals and a stipend from APA Publishing for her role as Deputy Editor at The American Journal of Psychiatry. All other authors report no financial or other relationships relevant to the subject of this manuscript.
