## [Peer Review File · Nature Communications]

Ketamine increases activity of a fronto-striatal projection that regulates compulsive behavior in SAPAP3 knockout miceREVIEWER COMMENTS

Reviewer #1 (Remarks to the Author):

This manuscript by Davis et al. demonstrates that the dmPFC-DMS projection modulates the expression of compulsive grooming in the SAPAP3 KO mouse model. The authors show that increased activity of this circuit, induced by acute ketamine treatment, is associated with reductions in compulsive grooming behavior in SAPAP3 KO mice (for 24-48h). They also show that inhibiting this circuit using optogenetics in WT mice permits grooming behavior. Furthermore, offset of grooming behavior is associated with an increase in activity of this circuit. Furthermore, optogenetic stimulation of this circuit also reduces compulsive grooming in SAPAP3 KO mice. Therefore, activity of the dmPFC-DMS projection appears to reduce habitual behaviors such as grooming. The experiments are very cohesive and the data support the authors' conclusions. The questions addressed are highly important and novel. Several minor issues could be corrected to improve the manuscript, and are listed below.

- 1) The authors should consider a title change. Ketamine increased peak amplitude in SAPAP3 KO, but not WT mice. So the title is not entirely accurate.
- 2) The term “rescued” is slightly confusing. It would be clearer to say that that ketamine “reduced” compulsive grooming behavior.
- 3) The claim that the SAPAP3 KO model is the “best validated rodent model of compulsive behavior” should be removed for several reasons. First, the human genetic sequencing studies in TTM/OCD used very small sample sizes, and such small genetic studies are no longer considered to likely be replicable. While OCD patients show reduced cognitive flexibility, they do not consistently show impaired reversal learning, as reported in SAPAP3 KO mice. Furthermore, SAPAP3 KO mice show reduced compulsive grooming after only subchronic SSRI treatment, which is inconsistent with the response in OCD patients. Numerous mouse KO models exhibit compulsive behaviors, and it is unclear which of these best models OCD compulsivity. We do not suggest that the SAPAP3 KO model is inappropriate for studying compulsive behavior, but suggest it is not the “best validated” model and this claim is very misleading. The field should remain open to the use of several different mouse models of OCD-like behavior, without considering others inferior to the SAPAP3 KO model.
- 4) A potential weakness of the paper is that grooming is the only compulsive behavior studied.
- 5) The 30mg/kg ketamine dose is somewhat high. Did the authors first try a dose response? Perhaps this would be worth mentioning or showing in the supplement?
- 6) On page 7, the authors say that the attenuation in KO grooming returned to baseline levels by day 7. But statistically it was back to normal on day 1 for duration and day 3 for bouts. This should be corrected. Even though the time to offset of the effect is slightly different in rodents and humans, the similarity is still impressive. This problem should also be corrected in the discussion.
- 7) On page 12 line 58, state what the animal model is.

8) On page 13, the authors refer to a trend following saline injection. If this is truly a trend ($p < .10$) the statistic should be given in the results and labelled on the figure. Otherwise, the word trend should be removed.

9) The authors suggest that ketamine reduces compulsive grooming in SAPAP3 KO mice by increasing activity of the dmPFC-DMS projection. The results are only consistent with this idea. More direct evidence would have shown that inhibiting this projection can prevent the effect of ketamine treatment in these mice.

Reviewer #2 (Remarks to the Author):

Davis et al use the Sapap3 KO mouse model of OCD to study how dmPFC-DMS projections participate in the control of grooming. They further explore how ketamine exerts an effect on this circuit, likely contributing to the therapeutic effect of the drug. The results are novel and interesting, and could contribute to translational work on the use of ketamine in OCD and how it could be effectively combined with brain stimulation therapies. Although I find the manuscript quite interesting, there are a number of issues that should be addressed prior to publication.

Major Comments:

1. The authors need to show histology for all their experiments. This is lacking throughout the paper. Readers need to see the injection sites, viral spread, implant placements. We need to know the extent to which dmPFC neurons are labeled (what %) and what layers. Are DMS-projecting PFC neurons projecting elsewhere as well? CAV-2 toxicity is sometimes a problem – is DMS intact after these injections?

2. The authors state there is a sustained decrease in dmPFC-DMS activity in WT mice during grooming. However, the duration of the decrease doesn't exactly match the average grooming bout duration given. I'm not sure if there is just a variability in the grooming bout duration that's difficult to see with this visualization. It could be helpful to show the offset of grooming PETHs with the traces normalized to a baseline pre-grooming bout. This way we could see if the activity immediately before the end of grooming is above or below the general non-grooming baseline.

3. Comparing Fig 1 and Fig 3, I'm not sure why the WT vs KO difference switches from a difference in peak frequency to no difference (but perhaps a trending decrease in peak amplitude instead). Is there a reason for this? Could it be related to the # of grooming bouts, given later grooming-specific differences shown? Also, given that it's actually dips in the signal that correlate with grooming, one might rather

expect to see some quantification of dips in the signal, which could be more consistently linked to genotype.

4. In Fig 3f, the authors compare signals in KO mice treated w/saline and ketamine, 1 day after injection. According to Fig 2, this should result in a decrease in grooming. But the average duration listed for grooming is actually higher in the KO-ketamine (12.21s) than in the KO-saline (9.92s). Is there a reason for this?

5. In Fig 3h+i, ketamine treatment seems to cause the signal in both groups to start going up before the end of grooming. I would be curious if the authors have a comment on this phenomenon (if it can be quantified).

6. Again in Fig 3 it would be nice to see some recordings of full grooming bouts. In the KO-ketamine group it looks like the signal goes quite high compared to baseline (3f), so the lack of an increase at the end of grooming could be due to an already high baseline, which would be interesting to report.

7. The recordings in Supp Fig 2 are only mildly helpful for interpretation, as the authors do not record specifically from DMS-projecting dmPFC cells. It is unclear to me which neurons they are patching e.g. what layer and how are the borders of dmPFC defined? I don't see any methods for electrophysiology, which limits my ability to evaluate these experiments' technical quality.

8. I'm concerned about 10min of continuous 5W green light. This amount of light, particularly green light, would be expected to induce significant heating (see Owen et al., Nat Neurosci, 2019). Long duration activations of NpHR have also been shown to alter chloride potential, with resulting effects on GABA transmission (Raimondo et al, Nat Neurosci, 2012). What is the reason the effect of inhibition is limited to the beginning of light delivery? The fading effect and/or the lack of time-locking to behavioral changes could be due to heating artifacts or changes in chloride potential. Given that the authors aren't taking advantage of the temporal precision of optogenetics here anyway, I would very much like to see them repeat these findings using another approach, such as chemogenetics.

9. In terms of translational potential, it would be very valuable to better characterize the duration and frequency of stimulation required for lasting effects on grooming, as well as how long the improvements last after the end of stimulation. It would also be very valuable to determine if the lasting effects of stimulation are due to plasticity of dmPFC-DMS synapses. Have the authors rescued the corticostriatal deficits previously reported in KO mice?

10. It would be great to see an experiment taking better advantage of the temporal precision of optogenetics. According to the author's theory, short optogenetic stimulation during a grooming bout should cause termination of the bout, perhaps with reasonably good temporal precision. It would be relatively easy and informative to test if this were true.

11. To better connect the ketamine and optogenetics findings, it would be good to demonstrate what effect ketamine treatment has on the efficacy of optogenetic stimulation. Are the effects of optogenetic stimulation occluded or facilitated by prior ketamine treatment?

Minor Comments:

1. Can the authors clarify what timeframes the AUCs are calculated over? I'm assuming it's within the PETH window shown, but can't tell for sure.
2. I don't understand Supp Fig 1 – why is there a lack of ketamine's effect on distance traveled in (a) but then a large effect in (b)? In (c) you seem to mean a 200% not a 2% change, so the y-axis is mislabeled.
3. Are the error bars given showing standard error? The legends don't say. In general, it would be better to use visualizations that can better display the individual variability observed. Sapap3 KO mice can be very variable in the severity of their presentation.
4. Authors report that both male and female mice were used. Can they report on whether or not there are sex differences observed? Were all groups equally balanced in the inclusion of males and females?
5. All mice were at least 120d but what is the upper range used and was this variable experiment-to-experiment? How much variability was observed in the overgrooming phenotype and were the KOs used displaying lesions due to overgrooming at this point?
6. Please report viral titers used.
7. 2 of 15 KO were classified as non-responders to ketamine and did not have their photometry data included in the analysis. However, it would be interesting to see if the lack of a behavioral effect was also observed as the lack of an effect on dmPFC activity.
8. The fiber photometry analysis is described only as "using custom MATLAB code," however it would be useful to provide more detail on how the data was normalized e.g. using the 405nm signal. Some examples of raw traces would be helpful. As is, it's a bit difficult to evaluate the technical quality of the fiber photometry experiments.

Reviewer #3 (Remarks to the Author):

General Comments

This article by Gwynne L. Davis and colleagues investigates the potential therapeutic effect of on OCD-like behaviors and try to decipher their underlying neurophysiological substrate along the fronto-striatal pathway. To this aim, they use a combination of experimental approaches combining behavioral assays in Sapap3 mouse model to assess the effect of ketamine on compulsive behaviors, calcium imaging to assess neuronal activity in dorsomedial prefrontal and striatal areas, and optogenetic manipulation of this fronto-striatal pathway in Sapap3-KO and WT animals. Their main claim is that ketamine can

significantly decrease compulsive behavior in this model with an increase of fronto-striatal activity as a possible mechanistic explanation.

This study is of obvious clinical interest in the context of reconsidering the use of such substance as a possible pharmacological indication in psychiatric disease. From a more fundamental scientific point of view, it also try to confirm and/or propose some insight in the pathophysiology of compulsive behaviors.

However, I have multiple concerns about the way the authors analyses and interpret some of their data. Indeed, clarifications need to be provided on several aspects.

Major comments:

1. In figure 1e/f, authors report that there is a significant dip in calcium signal in WT mice not visible in Sapap3-KO mice when grooming starts. This could be an interesting marker but it is very odd that calcium activity decreases just before grooming (about 2 sec) in Sapap3 animals. It looks like the temporal realignment is somehow shifted in sapap3-ko mice, otherwise how could the author explain this pregrooming decrease?
2. Related to first comment, the significant effect observed in figure 1g seems to come from the wrong realignment of the data. It seems that the significant decrease would also appears in sapap3-ko mice if calcium data were realigned 2 sec before (when the deep occurs).
3. How do the authors explain the fact that in both groups, locomotor activity was decreased compare to baseline condition in KO and WT, and this over 7 days? It is important to assess correctly this parameter since it could be a confounding factor of a general decrease of activity.
4. Even if the grooming drastically reduce after 1hour, and still significantly after 1 day, this effect disappears after that (even if a trend is visible). Therefore, the authors should not over claim in their conclusion that there is an effect lasting up to 7 days, as observed in humans. According to their data, the effect last one day.
5. How does the data from figure3c recompile with those from figure 1d? It looks like the original peak frequency difference between WT and KO is now absent, even in saline conditions?
6. In the same, way, I have difficulties understanding the rationale of choosing different durations to compare AUC between figure1 and figure 3e-f. I actually couldn't find on which duration was calculated AUC in figure 3g, I assume on the entire grooming episode? The authors should better explain their motivation to do so.
7. Figure 4 is somehow easier to follow and bring some convincing causal explanation of the implication of dmPFC-striatal pathway in the regulation of grooming. The effect they observe with the inhibition of this pathway are rather new and interesting. Their fronto-striatal optogenetic-induced excitation approach is very similar to two previous studies manipulating also important cortico-striatal circuits in the context of compulsive behaviours. In fact, their effect could seem somehow opposite to what have been observed in Ahmari et al. publication (2013) where they manipulated the mOFC-ventral striatum pathway but more similar to Burguiere et al. (2013) results where the authors stimulated the IOFC-DMS

pathway. The authors should better discuss these previous results and the possible complementary roles of these different circuits in compulsive behaviours in their discussion.

8. The last panel of figure 4 on chronic effect of repeated stimulation is very interesting and somehow a bit buried under all the other results. Again this result should be discussed in regard of Ahmari et al. study where they induce the opposite effect with a similar protocol.

9. I understand that the ketamine effect observed on dmPFC activity is not straightforward to interpret and the explanations of the authors are reasonable, explaining that dmPFC-increased activity in Sapap3-KO mice may be a compensatory mechanism to counteract hyperactive downstream striatal areas (p.13, l.2-3). However, this idea is not properly illustrated in figure 5 where DMS activity seems to increase with ketamine or optogenetic stimulation. If I follow their reasoning, you would expect a decrease of DMS activity? This cortical regulation of striatal activity has been proposed as a mechanistic possibility through feed-forward inhibition mechanism and could be discussed.

Minor comments:

1. In Figure 1d, authors report an increased frequency of calcium transient peaks in DMPFC. How do the authors interpret this fiberphotometry readout compared to more direct population recording observed with extracellular electrophysiology? More specifically in this model, at least two previous studies did not find difference of neuronal activity (in terms of firing rates) in a nearby cortical structure (OFC), could the author better discuss that? Did the authors perform any ephys recording in dmPFC to better interpret and compare with their fiberphotometry data?

2. In suppl figure 1a/b, I do not understand how distance travelled was measured (and could not find it in the methods), was it always a 10min time window at different time points?

3. The number of mice used is not systematically reported and sometimes difficult to spot, e.g. figure 2, suppl fig1, ..., please systematically indicate the "N=" on the figures or results section.

4. In figure 3e (up), the Y-axis legend needs to be corrected to "Z-score"

RESPONSE TO REVIEWERS

We would like to thank the reviewers for their helpful feedback on our original submission. We were encouraged to see that they found our study “highly important,” “novel and interesting,” “impressive,” and appreciated its translational relevance. They also identified some important concerns and suggested additional experiments and analyses that we believe have greatly strengthened the manuscript. Some of the more substantial additions include: 1) closed-loop optogenetic stimulation and inhibition experiments, capitalizing upon the temporal precision of optogenetics to show that we can causally extend or truncate individual grooming bouts. 2) an entirely new figure combining our optogenetic and pharmacological (ketamine) effects to better link these two findings, and 3) updating our fiber photometry analysis to better visualize the data and improve clarity. We have addressed these concerns and others brought forth, as detailed below in response to individual comments (responses in blue text).

Reviewer #1 (Remarks to the Author):

This manuscript by Davis et al. demonstrates that the dmPFC-DMS projection modulates the expression of compulsive grooming in the SAPAP3 KO mouse model. The authors show that increased activity of this circuit, induced by acute ketamine treatment, is associated with reductions in compulsive grooming behavior in SAPAP3 KO mice (for 24-48h). They also show that inhibiting this circuit using optogenetics in WT mice permits grooming behavior. Furthermore, offset of grooming behavior is associated with an increase in activity of this circuit. Furthermore, optogenetic stimulation of this circuit also reduces compulsive grooming in SAPAP3 KO mice. Therefore, activity of the dmPFC-DMS projection appears to reduce habitual behaviors such as grooming. The experiments are very cohesive and the data support the authors' conclusions. The questions addressed are highly important and novel. Several minor issues could be corrected to improve the manuscript, and are listed below.

1) The authors should consider a title change. Ketamine increased peak amplitude in SAPAP3 KO, but not WT mice. So the title is not entirely accurate.

Thank you for this feedback on the title. We agree that the previous title implied some overgeneralization, and it has now been changed to be more specific to KO mice as follows: “Ketamine increases activity of a fronto-striatal projection that regulates compulsive behavior in SAPAP3 KO mice.”

2) The term “rescued” is slightly confusing. It would be clearer to say that that ketamine “reduced” compulsive grooming behavior.

The four places where we previously said “rescued” have been changed to “reduced” as suggested (page 4, line 41, page 7, line 128, page 9, line 172, and page 14, line 305).

3) The claim that the SAPAP3 KO model is the “best validated rodent model of compulsive behavior” should be removed for several reasons. First, the human genetic sequencing studies in TTM/OCD used very small sample sizes, and such small genetic studies are no longer considered to likely be replicable. While OCD patients show reduced cognitive flexibility, they do not consistently show impaired reversal learning, as reported in SAPAP3 KO mice. Furthermore, SAPAP3 KO mice show reduced compulsive grooming after only subchronic SSRI treatment, which is inconsistent with the response in OCD patients. Numerous mouse KO models exhibit compulsive behaviors, and it is unclear which of these best models OCD compulsivity. We do not suggest that the SAPAP3 KO model is inappropriate for studying compulsive behavior, but suggest it is not the “best validated” model and this claim is very misleading. The field should remain open to the use of several different mouse models of OCD-like behavior, without considering others inferior to the SAPAP3 KO model.

We thank the reviewer for this feedback. We certainly did not mean to indicate that this model is superior to all others. We agree that as previously written, this claim is misleading, and we recognize the importance of using a variety of models in understanding complex psychiatric conditions. Our goal was to acknowledge the substantial research and groundwork others have done demonstrating the importance and validity of this model for understanding compulsive behavior. Therefore, we have now changed the text to read: "One well validated rodent model."

4) A potential weakness of the paper is that grooming is the only compulsive behavior studied.

We agree that ultimately it would be powerful to demonstrate that ketamine provides a therapeutic effect in another assay for compulsive behavior. We are actively trying to identify an alternative task that correlates with compulsive grooming, as other work looking at operant assays (such as reversal learning) relevant to OCD have showed that reversal learning does not correlate with compulsive grooming in SAPAP3 KO mice (Manning et al. 2019). We have piloted some contingency degradation tasks and similarly not had much luck. We look forward to identifying and validating another behavioral measure in the SAPAP3 KO mice that correlates with their compulsive grooming phenotype, although this is unfortunately beyond the scope of the current paper. Until we can identify such a task to test with ketamine, we hope that the reviewer appreciates the expansion of work we have done with the grooming phenotype in the revision of our manuscript. We have another manuscript from the laboratory that will be submitted shortly for peer review examining other behavioral symptom domains in SAPAP3 KO mice, such as altered valence processing, relevant to OCD.

Manning, E. E., Dombrowski, A. Y., Torregrossa, M. M., & Ahmari, S. E. (2019). Impaired instrumental reversal learning is associated with increased medial prefrontal cortex activity in Sapap3 knockout mouse model of compulsive behavior. *Neuropsychopharmacology*, 44(8), 1494–1504.
<https://doi.org/10.1038/s41386-018-0307-2>.

5) The 30mg/kg ketamine dose is somewhat high. Did the authors first try a dose response? Perhaps this would be worth mentioning or showing in the supplement?

We did indeed try a lower dose in the beginning (20 mg/kg) in a small group of animals. At this dose we did not see any therapeutic effect. We have now included this data in the new Supplemental Figure 12. We have also added a statement and citation in our methods section explaining our dose choice (page 23, lines 565-567), which was based off of work from Dr. Todd Gould's lab investigating the rapid anti-depressant effects of ketamine in mice.

6) On page 7, the authors say that the attenuation in KO grooming returned to baseline levels by day 7. But statistically it was back to normal on day 1 for duration and day 3 for bouts. This should be corrected. Even though the time to offset of the effect is slightly different in rodents and humans, the similarity is still impressive. This problem should also be corrected in the discussion.

This is absolutely correct and was poorly worded in our initial text. We have corrected this in our results and discussion sections when talking about the behavioral pharmacology experiments (page 7, line 129 and page 17, line 383). These corrections are now highlighted in the updated manuscript.

7) On page 12 line 58, state what the animal model is.

We have fixed this problem so that now it explicitly states that the model is the SAPAP3 KO mouse (page 17, line 394). We thank the reviewer for alerting us that this was not clear in the manuscript.

8) On page 13, the authors refer to a trend following saline injection. If this is truly a trend ($p < .10$) the statistic should be given in the results and labelled on the figure. Otherwise, the word trend should be removed.

We would like to thank the reviewer for their input on this point. Because we use a 2-way repeated measures ANOVA to analyze the data it limits our multiple comparisons analysis to either comparing means across rows or across columns, not allowing for comparison of means regardless of row or column. As such, in our main figure the multiple comparisons are represented using a Tukey's test comparing means across experimental groups within a day. However, in response to this comment, we went back and re-ran the statistical analysis on the data and set the multiple comparisons to assess within an experimental group across days instead. When we do this, we see that saline injection causes a significant increase in grooming duration from baseline in KO mice 24 hours post-injection. We have now included a new **Supplementary Figure 4** to demarcate the trends and points of significance with this alternative analysis. We have also adjusted the discussion text accordingly to reflect the updated statistical information as well (page 19, line 442). We find this effect of saline injection on KO grooming levels intriguing, and it has inspired a separate project in the lab which has generated some interesting preliminary findings about altered stress responses in SAPAP3 KO mice.

9) The authors suggest that ketamine reduces compulsive grooming in SAPAP3 KO mice by increasing activity of the dmPFC-DMS projection. The results are only consistent with this idea. More direct evidence would have shown that inhibiting this projection can prevent the effect of ketamine treatment in these mice.

We completely agree that in the initial submission these data (the reduction in grooming and the increased activity of the dmPFC-DMS projection) were simply consistent and not directly linked. As the other reviewers and editor also noted, more directly linking these findings will greatly strengthen the manuscript. Therefore, we have now added an entirely new main figure (**Figure 6**) addressing this point where we combine optogenetic inhibition with ketamine administration in KO mice. This new dataset has greatly contributed to our understanding of the fiber photometry data. KO-NpHR mice increase their grooming further when we inhibit the dmPFC-DMS circuit supporting our hypothesis that the increased dmPFC-DMS activity in our baseline fiber photometry recordings is a compensatory mechanism. Additionally, we were able to block this optogenetically-induced increase in grooming in the KO mice with ketamine injection given 24 hours prior to circuit manipulation. This result supports our hypothesis that ketamine is exerting its therapeutic effect on compulsive grooming by increasing the dmPFC-DMS activity past some critical threshold.

Reviewer #2 (Remarks to the Author):

Davis et al use the Sapap3 KO mouse model of OCD to study how dmPFC-DMS projections participate in the control of grooming. They further explore how ketamine exerts an effect on this circuit, likely contributing to the therapeutic effect of the drug. The results are novel and interesting, and could contribute to translational work on the use of ketamine in OCD and how it could be effectively combined with brain stimulation therapies. Although I find the manuscript quite interesting, there are a number of issues that should be addressed prior to publication.

Major Comments:

1. The authors need to show histology for all their experiments. This is lacking throughout the paper. Readers need to see the injection sites, viral spread, implant placements. We need to know the extent to which dmPFC neurons are labeled (what %) and what layers. Are DMS-projecting PFC neurons projecting elsewhere as well? CAV-2 toxicity is sometimes a problem – is DMS intact after these injections?

This was an oversight on our part, and have now included detailed histology figures showing injection and implantation sites for all of our cohorts as well as example images to visualize viral spread (**Supplementary Figures 1, 7, and 8**). The size and location of the implants covers portions of layers 2/3 and 5 (as seen in the representative image). We observed viral expression across superficial and deep cortical layers consistent with other publications from our lab examining this same projection (Loewke et al. 2021)

Loewke AC, Minerva AR, Nelson AB, Kreitzer AC, Gunaydin LA. Fronto-striatal projections regulate innate avoidance behavior. *J Neurosci*. 2021 May 12;JN-RM-2581-20. doi: 10.1523/JNEUROSCI.2581-20.2021. Epub ahead of print. PMID: 34001628.

To address what % of dmPFC neurons are labeled, we calculated a ratio of GCaMP-positive cells to DAPI-labeled cells and saw an average labeling of 9.4%. We also counted GCaMP-positive cells in a 400-micron by 400-micron area directly under the implant site and saw that on average 45.5 neurons +/- 10 s.e.m. were labeled in this area. We have updated the methods text to include this information about cell counts (page 22, lines 534-538).

Regarding CAV-2 toxicity, the DMS appears to be intact except for some expected damage from the needle at the injection site. We had trouble finding literature in which CAV-2 toxicity was an issue, only finding articles that lauded CAV-2's low toxicity (Lavoie et al. 2020). A previous study has performed a TUNEL assay to assess toxicity in the striatum following injection of CAV2-Cre virus and found no difference in the number of apoptotic cells compared to controls, using equivalent viral titers to ours (4.1×10^{12} pp/mL versus 5×10^{12} pp/mL for our working dilution) (Li et al. 2018). These studies provide evidence that the CAV-2 virus, at a titer similar to ours and in the same brain region, did not cause any appreciable toxicity in the dorsal striatum.

Lavoie, A., Liu, B., (2020). Canine Adenovirus 2: A Natural Choice for Brain Circuit Dissection. *Frontiers in Molecular Neuroscience*, 13(9). <https://doi.org/10.3389/fnmol.2020.00009>.

Li SJ, Vaughan A, Sturgill JF, Kepecs A. A Viral Receptor Complementation Strategy to Overcome CAV-2 Tropism for Efficient Retrograde Targeting of Neurons. *Neuron*. 2018 Jun 6;98(5):905-917.e5. doi: 10.1016/j.neuron.2018.05.028. PMID: 29879392.

2. The authors state there is a sustained decrease in dmPFC-DMS activity in WT mice during grooming. However, the duration of the decrease doesn't exactly match the average grooming bout duration given. I'm not sure if there is just a variability in the grooming bout duration that's difficult to see with this visualization. It could be helpful to show the offset of grooming PETHs with the traces normalized to a baseline pre-grooming bout. This way we could see if the activity immediately before the end of grooming is above or below the general non-grooming baseline.

We thank the reviewer for this suggestion as this new way of visualizing the photometry data greatly clarifies our data compared to the initial submission. The grooming offset PETHs are now normalized to the same 5 second baseline window that was used for the grooming onset PETHs, which makes the AUC bar plots look even more striking (**Figures 1 and 3**). Additionally, we have added a new **Supplementary Figure 2** with several examples of raw traces during grooming bouts to further support and clarify these data.

3. Comparing Fig 1 and Fig 3, I'm not sure why the WT vs KO difference switches from a difference in peak frequency to no difference (but perhaps a trending decrease in peak amplitude instead). Is there a reason for this? Could it be related to the # of grooming bouts, given later grooming-specific differences shown? Also, given that it's actually dips in the signal that correlate with grooming, one might rather expect to see some quantification of dips in the signal, which could be more consistently linked to genotype.

We acknowledge the desire to compare baseline (uninjected) data to post-injection data. However, we think that it is reasonable that the differences observed in the baseline recordings are not the same as seen post-injection, given that the injection itself could have an effect. As such, we want to be cautious in comparing baseline values to post-injection values head-to-head in the manuscript. However, for the purpose of addressing this comment, we did go back and compare the averages at baseline versus post-injection. For WT mice, the average peak frequency does go up a little for both saline and ketamine injection compared to

baseline. This increase does not reach significance, but is trending that way (ordinary one-way ANOVA $P = 0.0877$). The amplitudes do go down slightly compared to baseline, but this again is not statistically significant (ordinary one-way ANOVA $P = 0.4898$). However, the trend in increased frequency in WT post-injection could explain why we no longer see a significant difference between WT and KO peak frequency in Figure 3. For KO, there is no difference in peak frequency across conditions (ordinary one-way ANOVA $P = 0.5634$), which indicates to us that the change in significance in peak frequency from Figure 1 to Figure 3 is being driven by the trend in increased peak frequency of WT animals post-injection. The average grooming epoch of WT animals post-injection is shorter than what it was at baseline by ~ 1 second. It could be that the increased peak frequency is the result of these shorter grooming epochs. This hypothesis is supported in part by our new KO closed-loop stimulation experiment (discussed in comment #10 below, **Figure 5b-d**), in which brief dmPFC-DMS projection stimulation only during grooming shortens the duration of grooming bouts.

Regarding quantification of “dips” in the photometry signal, these are quantified in our AUC analysis and it indeed appears to be strongly linked to genotype (e.g., **Figure 1g**). We hope the new visualization of the AUC data in our revised manuscript (in response to comment #2 above) will help clarify this.

4. In Fig 3f, the authors compare signals in KO mice treated w/saline and ketamine, 1 day after injection. According to Fig 2, this should result in a decrease in grooming. But the average duration listed for grooming is actually higher in the KO-ketamine (12.21s) than in the KO-saline (9.92s). Is there a reason for this?

This is a good observation and we understand how this discrepancy may cause some confusion. First of all, we want to highlight that ketamine does indeed decrease overall grooming as expected for the fiber photometry implanted KO mice, illustrated in Figure 3b (similar to the unimplanted mice in Figure 2). However, for generating the PETHs there are a couple of points to keep in mind: 1) all grooming epochs less than 3 seconds are filtered out of the photometry analysis and 2) in order to analyze a grooming epoch, the preceding 5 seconds must not contain grooming in it from a preceding grooming epoch so that the 5 second baseline can be generated. So, while ketamine does decrease grooming overall for the KO mice in Figure 3, the epochs available for analysis after the filtering criteria result in a slightly longer average.

5. In Fig 3h+i, ketamine treatment seems to cause the signal in both groups to start going up before the end of grooming. I would be curious if the authors have a comment on this phenomenon (if it can be quantified).

Using the same pre-groom baselines as for the groom-start PETHs (as a part of re-analysis to address comment #2 above), we no longer observe this phenomenon in the groom-end PETHs. This indicates to us that this rise was likely some sort of artifact generated by our previous baselining method for the groom-end PETHs. For the WT-ketamine mice you will now see that with the end of grooming there is a sharp increase in neural activity that is in line with the behavioral transition. There may still be the appearance of some increase in activity, occurring a little bit before the transition. However, when we quantify the mean signal in the 2 seconds preceding the end of grooming, there is no statistical difference between WT-saline and WT-ketamine mice (two-tailed t-test $P = 0.5004$). For the KO-ketamine mice, there is an increase in neural activity during grooming that is maintained immediately post-grooming, but the activity during grooming does not seem to ramp up the way it appeared with the previous baselining method. Instead, it appears to be more of a general increase across the board in neural activity during grooming, that is then maintained after grooming has ended. Below are some raw traces to highlight the differences between KO-ketamine and KO-saline animals in individual grooming bouts. As can be seen in these traces below and in the groom-end PETHs (**Figure 3**), the KO-ketamine mice display a lot more variability, but overall display an increase in activity.

Marker for borders of grooming onset and offset →

6. Again in Fig 3 it would be nice to see some recordings of full grooming bouts. In the KO-ketamine group it looks like the signal goes quite high compared to baseline (3f), so the lack of an increase at the end of grooming could be due to an already high baseline, which would be interesting to report.

We thank the reviewer for their input here, and ask them to please refer to our response to comment #5 as we believe it also addresses comment #6.

7. The recordings in Supp Fig 2 are only mildly helpful for interpretation, as the authors do not record specifically from DMS-projecting dmPFC cells. It is unclear to me which neurons they are patching e.g. what layer and how are the borders of dmPFC defined? I don't see any methods for electrophysiology, which limits my ability to evaluate these experiments' technical quality.

We agree with the reviewer that it would have been better to have projection-specific slice recordings from dmPFC-DMS projection neurons. Unfortunately, we no longer have a slice physiologist in the lab, limiting which experiments we could do. However, we did find a collaborator who kindly agreed to do some additional and projection-specific slice experiments for us. For these experiments, we looked at the release properties of dmPFC projection neurons in the DMS following ketamine and saline injection, to test whether ketamine altered pre- or post-synaptic properties. For both our original slice data and these new data, the methods are located at the end of the supplement in a Supplemental Methods section. Ketamine caused no difference in release probability as measured by PPR between KO-saline and KO-ketamine mice. Ketamine did abolish the difference in frequency of evoked asynchronous EPSCs seen between WT and KO mice, indicating that ketamine may increase dmPFC projection activity independent of mechanisms that affect release probability. The results and discussion sections have also been updated to reflect this new data (page 11, lines 219-237 and page 17, lines 403-427). The methods for our original data have been extended to include additional information to address questions about layers and how the borders were defined.

8. I'm concerned about 10min of continuous 5W green light. This amount of light, particularly green light, would be expected to induce significant heating (see Owen et al., Nat Neurosci, 2019). Long duration activations of NpHR have also been shown to alter chloride potential, with resulting effects on GABA transmission (Raimondo et al, Nat Neurosci, 2012). What is the reason the effect of inhibition is limited to the beginning of light delivery? The fading effect and/or the lack of time-locking to behavioral changes could be due to heating artifacts or changes in chloride potential. Given that the authors aren't taking advantage of the temporal precision of optogenetics here anyway, I would very much like to see them repeat these finding using another approach, such as chemogenetics.

We would like to thank the reviewer for their insight in this manner, and agree that altered chloride potential is a possible concern with continuous laser exposure. We attempted to address this concern with two new experiments 1) using a chemogenetic approach as suggested, and 2) by taking advantage of the temporal precision of optogenetics to perform a closed-loop inhibition experiment. The results of our chemogenetics experiment are discussed below, but we did not include them in the revised manuscript due to concerns detailed below. However, the temporally precise closed-loop inhibition experiment was successful and is now included in the new main **Figure 4**. Briefly, we were able to increase the average length of individual grooming bouts by turning the laser on at the start of grooming and turning it off when grooming stopped. Details about the results and methods of this experiment can be found in the revised manuscript (page 12, lines 264-279 and page 24, lines 582-586 and 591-593). We hope that the reviewer will be as excited about these results as we are.

For the chemogenetic experiment, we injected an inhibitory DIO-DREADD (hM4D) into the dmPFC and CAV2-Cre into the DMS. After a 5-week recovery time we then administered either saline or CNO via an i.p. injection at 2 different doses (1.5 and 3.0 mg/kg) and measured grooming at multiple different timepoints. All animals received both a saline injection and a CNO injection so that we could compare grooming differences within animals. Animals were balanced across days for receiving saline or CNO first. At least 2 days were given between trying different timepoints to allow for washout. Unfortunately, this experiment was not successful, despite good viral expression in dmPFC-DMS projection neurons (example image directly below). Below you'll find a table summarizing our efforts (after image).

Cohort 1	Observation timepoint post-injection	Length of observation	Effect on grooming behavior	# of males	# of females	Virus
1.5 mg/kg	30 min	10 min	Trend towards decreased grooming in males	8	5	All mice injected with AAV5.hSyn.DIO.hM4D(Gi).mCherry in the dmPFC and CAV2-Cre in the DMS
3.0 mg/kg	10 min	10 min	A significant increase in grooming during the second 5 min block. Collected on the same day as the 60 min time point			
	5 min	10 min	No effect observed, increase in			

			grooming starting 5 min post injection from the 10 min post-injection time point not replicated. Collected on the same day as the 2.5 hour time point.			
	60 min	10 min	No effect observed, collected on the same day as the 10 min time point			
	2.5 hours	10 min	Trend towards increased grooming (P = 0.09), however this is solidly in the time range where clozapine could be causing an effect. Collected on the same day as the 5 min time point.			
Cohort 2	Observation timepoint post-injection	Length of observation	Effect on grooming behavior	# of males	# of females	Virus
3.0 mg/kg	0 min	20 min	Tried to confirm the 5-10 min behavior in cohort 1 and hopefully catch more data with the longer time frame that should be during CNO's peak bioavailability in the brain. No effect observed	6 hM4D 5 eYFP	6 hM4D 6 eYFP	Half of the mice were injected with AAV5.hSyn.DIO.hM4 D(Gi).mCherry in the dmPFC and CAV2-Cre in the DMS. Half of the mice were injected with AAV5.EF1a-DIO.eYFP in the dmPFC and CAV2-Cre in the DMS.

It is unclear why the experiment was unsuccessful, though we have concerns about the effectiveness of the CNO (Sigma, SML2304-5MG). These concerns include:

1. Reports/concerns that CNO does not cross the blood brain barrier very well, requiring cannulation for brain region specific administration. Though we did find a paper that did assess timepoints when CNO was at its highest in brain tissue post-injection and did try several trials using those timepoints.

Jendryka, M., Palchadhuri, M., Ursu, D., van der Veen, B., Liss, B., Kätzel, D., Nissen, W., & Pekcec, A. (2019). Pharmacokinetic and pharmacodynamic actions of clozapine-N-oxide, clozapine, and

compound 21 in DREADD-based chemogenetics in mice. *Scientific Reports*, 9(1), 4522. <https://doi.org/10.1038/s41598-019-41088-2>.

2. Timepoints often used for CNO are actually after CNO has been reverse metabolized into clozapine. A recent study demonstrated that mice were trained in a lever-pressing clozapine discrimination task and successfully discriminated at 30 minutes after CNO was administered subcutaneously at doses as low as 1.25 mg/kg.

Manvich, D.F., Webster, K.A., Foster, S.L. *et al.* The DREADD agonist clozapine *N*-oxide (CNO) is reverse-metabolized to clozapine and produces clozapine-like interoceptive stimulus effects in rats and mice. *Sci Rep* 8, 3840 (2018). <https://doi.org/10.1038/s41598-018-22116-z>.

We did try timepoints where, in theory, it would be clozapine interacting with the DREADDs, however, we did not see any effect. Interestingly, there is some evidence that chronic administration of clozapine itself can cause increased grooming:

Kang, S., Noh, H. J., Bae, S. H., Kim, Y. S., Lew, H., Lim, J., Kim, S. J., Hong, K. S., Rah, J. C., & Kim, C. H. (2020). Clozapine generates obsessive compulsive disorder-like behavior in mice. *Molecular brain*, 13(1), 84. <https://doi.org/10.1186/s13041-020-00621-5>.

3. We attempted a positive control experiment for CNO in which we injected mice with inhibitory DIO-DREADDs in M2 cortex and CAV2-Cre into the striatum to try to decrease locomotion, as optogenetic stimulation of this pathway promotes locomotor behavior (Magno et al. 2019). Once mice had recovered from surgery (5 weeks) we injected them either with 3mg/kg CNO or saline and placed them in an open field for 60 minutes post injection. Unfortunately, we again saw no effect of CNO, although it was only in three mice. After all of these negative DREADD data, we decided our efforts were best spent on conducting temporally-precise optogenetic inhibition instead, which was far more successful (as discussed above and in the new **Figure 4**).

Magno, L., Tenza-Ferrer, H., Collodetti, M., Aguiar, M., Rodrigues, A., da Silva, R. S., Silva, J., Nicolau, N. F., Rosa, D., Birbrair, A., Miranda, D. M., & Romano-Silva, M. A. (2019). Optogenetic Stimulation of the M2 Cortex Reverts Motor Dysfunction in a Mouse Model of Parkinson's Disease. *The Journal of neuroscience : the official journal of the Society for Neuroscience*, 39(17), 3234–3248. <https://doi.org/10.1523/JNEUROSCI.2277-18.2019>.

9. In terms of translational potential, it would be very valuable to better characterize the duration and frequency of stimulation required for lasting effects on grooming, as well as how long the improvements last after the end of stimulation. It would also be very valuable to determine if the lasting effects of stimulation are due to plasticity of dmPFC-DMS synapses. Have the authors rescued the corticostriatal deficits previously reported in KO mice?

We agree that understanding the plasticity at dmPFC-DMS synapses that produces a lasting reduction in grooming behavior will be a valuable contribution to the field. We attempted a second chronic stimulation experiment using different parameters to better characterize the duration and frequency required for lasting effects on grooming, which unfortunately yielded negative data. As detailed below, given the number of mice and timeframe needed to cover this large parameter space, we believe that a thorough characterization of these parameters and related plasticity is beyond the scope of this present manuscript.

As reported (and clarified) in the manuscript, our chronic reduction in grooming was evident after the mice had received a total of 4 different stimulation sessions that occurred in 3 different environmental contexts with rest days in between. The acute stimulation effects that we report are from the first stimulation session. Because

our chronic stimulation effect was produced by stimulating across different environments, we tried a different stimulation protocol similar to Ahmari et al. (*Science*, 2013). For each session, a mouse underwent a 5 minute baseline, 5 minute laser on, 5 minute laser off, 5 min laser on, and 5 min laser off cycle (25 minutes total, with 10 minutes of stimulation) in the cylinder we used for our other grooming measurements. We repeated this schedule for 5 days so that the mice received 50 minutes of laser stimulation total, an equivalent amount to what they received that produced the effects in the new **Figure 5h and 5i**. Unfortunately, we saw no effect on grooming compared to baseline using this type of stimulation protocol. We also tried doing a longer session consisting of 20 minutes of stimulation. We then looked at grooming behavior 24 hours and 72 hours later and saw no effect (see graph below, 2-way RM ANOVA: interaction $P = 0.8182$, day $P = 0.2780$, virus $P = 0.6771$, subject $P = 0.0006$, eYFP KO = 7, ChR2 KO = 8). It is interesting that 5 straight days of stimulation in the same environment does not result in a chronic rescue effect. This could indicate that the chronic reduction in grooming behavior is the result of having stimulation sessions interspersed by rest days, such that 5 days in a row doesn't produce the same plasticity, or possibly that stimulation sessions in different environments drives dmPFC-DMS changes that ultimately lead to a sustained decrease in grooming. However, a comprehensive study of which types of stimulation protocols produce a lasting effect and how they affect the plasticity of dmPFC-DMS synapses is, we believe, beyond the scope of this paper.

Ahmari, S. E., Spellman, T., Douglass, N. L., Kheirbek, M. A., Simpson, H. B., Deisseroth, K., Gordon, J. A., & Hen, R. (2013). Repeated cortico-striatal stimulation generates persistent OCD-like behavior. *Science* (New York, N.Y.), 340(6137), 1234–1239. <https://doi.org/10.1126/science.1234733>.

10. It would be great to see an experiment taking better advantage of the temporal precision of optogenetics. According to the author's theory, short optogenetic stimulation during a grooming bout should cause termination of the bout, perhaps with reasonably good temporal precision. It would be relatively easy and informative to test if this were true.

We would like to thank the reviewer for this suggestion. We ended up doing two separate experiments taking better advantage of the temporal precision of optogenetics. One of these experiments has already been discussed in comment #8 above (closed-loop inhibition, which extended individual grooming bouts). To address this comment, we also a closed-loop stimulation experiment in KO mice to address whether brief stimulation could interrupt and terminate individual grooming bouts. Briefly, we were able to decrease the average length of individual grooming epochs by turning the laser on at the start of grooming and turning it off when grooming had stopped. Details about the results (**Figure 5d**) and methods of this experiment can be

found in the revised manuscript (page 24, lines 582-586 and page 13, lines 286-299). This brief stimulation shortened the grooming epoch but did not immediately stop the grooming behavior. We think that this could be for several reasons. One possibility is that since the stimulation is briefer in this closed-loop experiment (lasting seconds versus minutes), bilateral stimulation may be necessary to achieve immediate termination of grooming behavior. However, to keep this experiment consistent with our initial ChR2 stimulation experiment, we only did unilateral stimulation and used the same power and stimulation parameters. Alternatively, as mentioned in our original text, we hypothesize that this circuit acts more in a modulatory or “permissive signaling” capacity, rather than directly driving the behavior, as mentioned in our discussion (page 19, lines 437-451).

11. To better connect the ketamine and optogenetics findings, it would be good to demonstrate what effect ketamine treatment has on the efficacy of optogenetic stimulation. Are the effects of optogenetic stimulation occluded or facilitated by prior ketamine treatment?

This point was also raised by Reviewer 1, and we agree that better connecting the ketamine and optogenetic findings has greatly strengthened our revised manuscript. As described in response to Reviewer 1 comment #9, we have now added an entirely new main figure (**Figure 6**) addressing this point where we combine optogenetic inhibition with ketamine administration in KO mice. This new dataset has greatly contributed to our understanding of the fiber photometry data. KO-NpHR mice increase their grooming further when we inhibit the dmPFC-DMS circuit supporting our hypothesis that the increased dmPFC-DMS activity in our baseline fiber photometry recordings is a compensatory mechanism. Additionally, we were able to block this optogenetically-induced increase in grooming in the KO mice with ketamine injection given 24 hours prior to circuit manipulation. This result supports our hypothesis that ketamine is exerting its therapeutic effect on compulsive grooming by increasing the dmPFC-DMS activity past some critical threshold. We hope that the reviewer is as excited by these data as we are.

Minor Comments:

1. Can the authors clarify what timeframes the AUCs are calculated over? I'm assuming it's within the PETH window shown, but can't tell for sure.

For each grooming event, an AUC is calculated for the duration of the grooming window. It is then divided by the grooming time length to produce a normalized AUC value so that we can average AUC data from grooming epochs of different lengths. We have added text to both the methods and the results section further clarifying this (line 88 and lines 625-626).

2. I don't understand Supp Fig 1 – why is there a lack of ketamine's effect on distance traveled in (a) but then a large effect in (b)? In (c) you seem to mean a 200% not a 2% change, so the y-axis is mislabeled.

We thank the reviewer for catching this oversight. We had accidentally left the values as ratios rather than percentages. The figure has now been updated it so that it correctly reflects % change.

3. Are the error bars given showing standard error? The legends don't say. In general, it would be better to use visualizations that can better display the individual variability observed. Sapap3 KO mice can be very variable in the severity of their presentation.

Error bars represent standard error of the mean (s.e.m.). We have added statements to both the statistical analysis section as well as on each figure legend indicating that data represents means +/- s.e.m (line 660). While we agree that being able to see individual variability is important, most our experiments have sample sizes of over ten per genotype, making the data difficult to read when presenting individual data points. The email we received from *Nature Communications* stated that “all data points should be shown for plots with a sample size less than 10.” Nevertheless, as you'll see in response to minor point #5 below, we have added

some supplemental figures addressing variability. Additionally, should our manuscript be accepted, we will provide the source data file containing all the raw values for each experiment (as recommended by *Nature Communications*), which will allow readers to look into the variability should they desire to.

4. Authors report that both male and female mice were used. Can they report on whether or not there are sex differences observed? Were all groups equally balanced in the inclusion of males and females?

We made an effort to keep cohorts equally balanced by sex, though various factors did affect sex distribution such as animals being removed from the dataset due to mistargeting, poor recovery post-surgery, and the stochastic nature of breeding. We have now included a table in the supplement (**Supplementary Figure 10**) showing the exact sex distribution for all experiments. Because our experiments were not intentionally designed to compare sexes, we do urge some caution when discussing post-hoc analysis of sex differences. However, to address this question, we did do some analysis on datasets where there were at least 4 animals per sex. In doing this, we saw no sex differences for WT-NpHR, WT-ChR2, or KO-ChR2 mice in response to optogenetic manipulation. We also saw no sex difference in grooming in WT or KO mice post-injection for both saline and ketamine. While there was not a main significant effect of sex for KO-saline mice, there was a significant interaction between post-injection time and sex, with female KO-saline mice showing higher levels of grooming 24 hours post-injection.

5. All mice were at least 120d but what is the upper range used and was this variable experiment-to-experiment? How much variability was observed in the overgrooming phenotype and were the KOs used displaying lesions due to overgrooming at this point?

In the same table where we have provided sex distribution, we have also provided the mean, median, and range of ages for each experiment (**Supplementary Figure 10**). Below this paragraph we provide a violin plot to illustrate the differences. Because WT mice are much more prolific in our colony, the ages of the mice used in the WT NpHR experiment (Figure 4) are much more tightly clustered than the other experiments involving SAPAP3 KO mice. As such, the mean age of the WT NpHR is considered significantly different compared to the other experiments when assessed with a Kruskal-Wallis test followed by a Dunn's multiple comparisons test (Kruskal-Wallis test $P < 0.0001$; Dunn's multiple comparisons $*P < 0.05$, $**P < 0.01$, $***P < 0.001$, $****P < 0.0001$). The KO NpHR cohort contains the oldest mice of all the experiments, as well as the highest mean age. This is in part because we had to use mice bred prior to COVID-19 mandated shutdowns of all new breeding that occurred in our facilities in late March 2019, which forced us to use mice older than we normally would in order to achieve sufficient sample sizes. Regarding grooming variability, as expected there is indeed some variability across mice. We assessed grooming duration in KO mice across experiments to determine if there were significant changes in grooming levels between experiments, and there were not (Kruskal-Wallis test $P = 0.0802$, violin plot below). The violin plot below showing variability in KO grooming duration across experimental cohorts is now also available in **Supplementary Figure 11** along with information about sample sizes and statistical analysis.

7. 2 of 15 KO were classified as non-responders to ketamine and did not have their photometry data included in the analysis. However, it would be interesting to see if the lack of a behavioral effect was also observed as the lack of an effect on dmPFC activity.

To address this comment, we went back and analyzed the photometry data of the non-responders. Interestingly, the non-responders did not have an increase in amplitude nor the normalized area under the curve that was observed in animals that behaviorally responded to ketamine. We have added these results to **Figure 3**.

8. The fiber photometry analysis is described only as “using custom MATLAB code,” however it would be useful to provide more detail on how the data was normalized e.g. using the 405nm signal. Some examples of raw traces would be helpful. As is, it’s a bit difficult to evaluate the technical quality of the fiber photometry experiments.

We agree and apologize for this oversight. **Supplementary Figure 2** now contains several example raw traces. We have also expanded our methods section to include the following: “Briefly, the 405 nm signal was regressed against the 470 nm signal and the polyfit MATLAB function (least squares method) was used to determine the coefficients to calculate a fitted 405 nm signal. The DF/F was then calculated ((490nm signal - fitted 405nm signal) / (fitted 405nm signal)). For peak amplitude and frequency calculations, we first detected all Ca²⁺ transient peaks throughout the signal using custom peak detection code using a running average method to calculate the peak to trough value. We used a 10 second trough window (window during convolution for finding running average trough) and a 1 second temporal window (minimum amount of time between peaks). Once peaks were detected, we then calculated the average frequency and amplitude of these peaks.”

Reviewer #3 (Remarks to the Author):

General Comments

This article by Gwynne L. Davis and colleagues investigates the potential therapeutic effect of on OCD-like behaviors and try to decipher their underlying neurophysiological substrate along the fronto-striatal pathway. To this aim, they use a combination of experimental approaches combining behavioral assays in Sapap3 mouse model to assess the effect of ketamine on compulsive behaviors, calcium imaging to assess neuronal activity in dorsomedial prefrontal and striatal areas, and optogenetic manipulation of this fronto-striatal pathway in Sapap3-KO and WT animals. Their main claim is that ketamine can significantly decrease compulsive behavior in this model with an increase of fronto-striatal activity as a possible mechanistic explanation. This study is of obvious clinical interest in the context of reconsidering the use of such substance as a possible pharmacological indication in psychiatric disease. From a more fundamental scientific point of view, it also try to confirm and/or propose some insight in the pathophysiology of compulsive behaviors. However, I have multiple concerns about the way the authors analyses and interpret some of their data. Indeed, clarifications need to be provided on several aspects.

Major comments:

1. In figure 1e/f, authors report that there is a significant dip in calcium signal in WT mice not visible in Sapap3-KO mice when grooming starts. This could be an interesting marker but it is very odd that calcium activity decreases just before grooming (about 2 sec) in Sapap3 animals. It looks like the temporal realignment is somehow shifted in sapap3-ko mice, otherwise how could the author explain this pregrooming decrease?

We also thought this slight dip ahead of grooming was interesting. We went back to our raw data and our MATLAB code to confirm the temporal alignment, which is correct as presented. It does seem like the KOs in general have more variable baseline activity compared to WTs. We suspect that the slight dip before grooming may be a byproduct of this increased variability and is not a robust or meaningful marker. We have provided some examples of raw traces in **Supplementary Figure 2** to highlight the increased variability seen in KO relative to WT mice.

2. Related to first comment, the significant effect observed in figure 1g seems to come from the wrong realignment of the data. It seems that the significant decrease would also appears in sapap3-ko mice if calcium data were realigned 2 sec before (when the deep occurs).

As discussed in response to comment #1 above, the alignment of the data has been double checked and is correct. However, in order to make our data clearer we have removed these plots looking at the +/- 2 second transition window and have streamlined our AUC analysis as recommended by Reviewer 2 (e.g., comment #2). You will now find that we focus only on the reduction in neural signal during the grooming event as a whole and the increase in the neural signal at the termination of grooming.

3. How do the authors explain the fact that in both groups, locomotor activity was decreased compare to baseline condition in KO and WT, and this over 7 days? It is important to assess correctly this parameter since it could be a confounding factor of a general decrease of activity.

We agree that it is always important to consider potential locomotor confounds. In this case, the higher locomotor activity on the baseline day is to be expected as mice often move more while exploring a novel environment. It is normal for this locomotion to habituate across days as the environment becomes less novel; this habituation will occur across all groups. However, what we consider important is that there is no difference in locomotion within a genotype across treatment conditions. Both KO-ketamine and KO-saline mice have the same locomotor activity, while KO-ketamine mice have a decrease in grooming behavior and KO-saline mice have an increase in grooming behavior. To us this demonstrates that ketamine's effect on grooming behavior is independent of locomotion. To further assuage any concerns, we have calculated Pearson correlation coefficients and significance for grooming and locomotor for each day across the groups. As shown in the table below, no significant correlations were observed.

Experimental Group	Pearson r	R squared	P value (two-tailed)
WT-Saline mice			
Baseline	-0.3661	0.1340	0.1980
1 hour	-0.1026	0.01053	0.7270
1 day	-0.1865	0.03479	0.5232
3 days	-0.4199	0.1763	0.1350
7 days	0.1401	0.01963	0.6328
WT-Ketamine mice			
Baseline	-0.3081	0.09491	0.3058
1 hour	-0.2169	0.04706	0.4765
1 day	0.3513	0.1234	0.2392
3 days	-0.001707	2.912e-006	0.9956
7 days	0.1150	0.01323	0.7083
KO-Saline mice			
Baseline	0.4588	0.2105	0.1557
1 hour	0.5672	0.3217	0.0688
1 day	0.2653	0.07039	0.4304
3 days	0.3122	0.09750	0.3499
7 days	0.4423	0.1956	0.1731
KO-Ketamine mice			
Baseline	-0.2704	0.07310	0.3954
1 hour	0.3642	0.1326	0.2445
1 day	-0.2803	0.07857	0.3775
3 days	-0.2214	0.04904	0.4891
7 days	-0.4059	0.1648	0.1905

4. Even if the grooming drastically reduce after 1hour, and still significantly after 1 day, this effect disappears after that (even if a trend is visible). Therefore, the authors should not over claim in their conclusion that there is an effect lasting up to 7 days, as observed in humans. According to their data, the effect last one day.

This is absolutely correct and was poorly worded in our initial text, as Reviewer 1 also noted. We have corrected this in our results and discussion sections when talking about the behavioral pharmacology experiments (page 7, line 129 and page 17, line 383).

5. How does the data from figure3c recompile with those from figure 1d? It looks like the original peak frequency difference between WT and KO is now absent, even in saline conditions?

We acknowledge the desire to compare baseline (uninjected) data to post-injection data. However, we think that it is reasonable that the differences observed in the baseline recordings are not the same as seen post-injection, given that the injection itself could have an effect. As such, we want to be cautious in comparing baseline values to post-injection values head-to-head in the manuscript. However, for the purpose of addressing this comment, we did go back and compare the averages at baseline versus post-injection. For WT mice, the average peak frequency does go up a little for both saline and ketamine injection compared to baseline. This increase does not reach significance, but is trending that way (ordinary one-way ANOVA $P = 0.0877$). The amplitudes do go down slightly compared to baseline, but this again is not statistically significant (ordinary one-way ANOVA $P = 0.4898$). However, the trend in increased frequency in WT post-injection could explain why we no longer see a significant difference between WT and KO peak frequency in Figure 3. For KO, there is no difference in peak frequency across conditions (ordinary one-way ANOVA $P = 0.5634$), which indicates to us that the change in significance in peak frequency from Figure 1 to Figure 3 is being driven by the trend in increased peak frequency of WT animals post-injection. The average grooming epoch of WT

animals post-injection is shorter than what it was at baseline by ~1 second. It could be that the increased peak frequency is the result of these shorter grooming epochs. This hypothesis is supported in part by our new KO closed-loop stimulation experiment (discussed in Reviewer 2 comment #10, **Figure 5b-d**), in which brief dmPFC-DMS projection stimulation only during grooming shortens the duration of grooming bouts.

Additionally, we would further caution against making comparisons involving KO-baseline to KO-saline values as the saline injection does have a behavioral effect, significantly increasing grooming behavior 24 hours post-injection (**Supplementary Figure 4**). Highlighting this behavioral difference, we see between baseline and post-saline, KO amplitudes are significantly different (ordinary one-way ANOVA $P < 0.0001$, Dunnett's multiple comparisons to assess KO-ketamine and KO-saline relative to KO-baseline amplitudes: KO-baseline vs KO-saline $P < 0.0001$, KO-baseline vs KO-ketamine $P = 0.0406$). Given these differences, we feel it's important to compare WT post-injection data with KO post-injection data.

6. In the same way, I have difficulties understanding the rationale of choosing different durations to compare AUC between figure 1 and figure 3e-f. I actually couldn't find on which duration was calculated AUC in figure 3g, I assume on the entire grooming episode? The authors should better explain their motivation to do so.

We now see that this way of analyzing and plotting the data was confusing. As discussed in response to comment #2 above as well as in Reviewer 2 minor comment #1, we have streamlined our AUC analysis for clarity and have updated the text to reflect these changes (line 88 and lines 625-626). For each grooming event, an AUC is calculated for the duration of the grooming window. It is then divided by the grooming time length to produce a normalized AUC value so that we can average AUC data from grooming epochs of different lengths.

7. Figure 4 is somehow easier to follow and bring some convincing causal explanation of the implication of dmPFC-striatal pathway in the regulation of grooming. The effect they observe with the inhibition of this pathway are rather new and interesting. Their fronto-striatal optogenetic-induced excitation approach is very similar to two previous studies manipulating also important cortico-striatal circuits in the context of compulsive behaviours. In fact, their effect could seem somehow opposite to what have been observed in Ahmari et al. publication (2013) where they manipulated the mOFC-ventral striatum pathway but more similar to Burguiere et al. (2013) results where the authors stimulated the IOFC-DMS pathway. The authors should better discuss these previous results and the possible complementary roles of these different circuits in compulsive behaviours in their discussion.

We thank the reviewer for this input, and have now added text to the discussion section to better address some of these points and comparison to related previous OFC studies (lines 472-477).

8. The last panel of figure 4 on chronical effect of repeated stimulation is very interesting and somehow a bit buried under all the other results. Again this result should be discuss in regard of Ahmari et al. study where they induce the opposite effect with a similar protocol.

We thank the reviewer for this input, and as with the previous comment, we have now added text to the discussion section to better address the chronic stimulation effect in the context of previous OFC studies (lines 472-477).

9. I understand that the ketamine effect observed on dmPFC activity is not straightforward to interpret and the explanations of the authors are reasonable, explaining that dmPFC-increased activity in Sapap3-KO mice may be a compensatory mechanism to counteract hyperactive downstream striatal areas (p.13, l.2-3). However, this idea is not properly illustrated in figure 5 where DMS activity seems to increase with ketamine or optogenetic stimulation. If I follow their reasoning, you would expect a decrease of DMS activity? This cortical

regulation of striatal activity has been proposed as a mechanistic possibility through feed-forward inhibition mechanism and could be discussed.

We thank the reviewer for this suggestion. We have updated our summary schematic (now **Figure 7**) to more accurately summarize our findings and improve clarity. Part of the changes to our summary figure include removing the spike patterns from the DMS. Our original intention was to indicate that dmPFC-DMS projections had increased glutamatergic release in the DMS under ChR2 optogenetic stimulation conditions. However, as the reviewer has pointed out, this was not a clear illustration and our results do not show whether this manipulation causes decreased or increased DMS activity downstream. In part, it depends on whether there are changes in the proportion of dmPFC projections that synapse onto fast-spiking interneurons (FSIs) versus medium spiny neurons (MSNs) in the SAPAP3 KO mice, and then how this is affected by the presence of ketamine. Other SAPAP3 KO mouse studies have shown that the loss of *Sapap3* effects cortical synapses into the striatum differently. For instance, the lateral OFC had reduced input onto MSNs with no change in the number of synapses onto FSIs in KO mice, but projections from the secondary motor cortex showed a 6-fold increase onto both FSIs and MSNs in KO mice (Corbit et al. 2019).

Corbit VL, Manning EE, Gittis AH, Ahmari SE. Strengthened Inputs from Secondary Motor Cortex to Striatum in a Mouse Model of Compulsive Behavior. *J Neurosci*. 2019 Apr 10;39(15):2965-2975. doi: 10.1523/JNEUROSCI.1728-18.2018. Epub 2019 Feb 8. PMID: 30737313; PMCID: PMC6462450).

Minor comments:

1. In Figure 1d, authors report an increased frequency of calcium transient peaks in DMPFC. How do the authors interpret this fiberphotometry readout compare to more direct population recording observed with extracellular electrophysiology? More specifically in this model, at least two previous studies did not find difference of neuronal activity (in terms of firing rates) in a nearby cortical structure (OFC), could the author better discuss that? Did the authors performed any ephys recording in dmPFC to better interpret and compare with their fiberphotometry data?

We have not performed any *in vivo* electrophysiology recordings. The lack of projection-specificity using traditional extracellular recordings would be difficult to correlate with our projection-specific fiber photometry data. In an attempt to further elucidate projection-specific mechanisms, we have performed slice physiology recordings in the dmPFC to determine if there were any changes in intrinsic cellular properties 24 hours after either saline or ketamine injection. No intrinsic differences were seen in dmPFC neurons, which led us to hypothesize that our fiber photometry results likely represented alter *in vivo* recruitment of those neurons during grooming behavior. We have since added to our original electrophysiology data in the revised manuscript by examining release properties of dmPFC projections in the DMS under ketamine and saline conditions, as discussed in more detail in Reviewer 2 comment #7. For both our original slice data and the new slice data, the methods are located at the end of the supplement in a **Supplementary Methods** section. The results and discussion sections have also been updated to reflect the additional data (page 11, lines 219-237 and page 17, lines 403-427).

Regarding the electrophysiology work that has been done in the OFC, Burguière et al. (*Science*, 2013) showed no differences in lateral OFC (lOFC) neuron firing between WT and KO. However, more recent studies have seen differences in IOFC neuronal activity. Lei et al (*Journal of Psychiatry and Neuroscience*, 2019) demonstrated increased interneuron activity and increased bursting activity in pyramidal neurons in the IOFC. This could be due to different *in vivo* electrophysiology recording techniques or when they recorded during their respective tasks. However, similar to our work, when IOFC-DMS terminals are optogenetically stimulated there is a reduction of compulsive behavior. In our updated discussion we do discuss the optogenetic effect and how it relates to our own work (lines 469-472).

Burguière, E., Monteiro, P., Feng, G., & Graybiel, A. M. (2013). Optogenetic stimulation of lateral orbitofronto-striatal pathway suppresses compulsive behaviors. *Science* (New York, N.Y.), 340(6137), 1243–1246. <https://doi.org/10.1126/science.1232380>.

Lei, H., Lai, J., Sun, X., Xu, Q., & Feng, G. (2019). Lateral orbitofrontal dysfunction in the Sapap3 knockout mouse model of obsessive–compulsive disorder. *Journal of psychiatry & neuroscience : JPN*, 44(2), 120–131. <https://doi.org/10.1503/jpn.180032>.

2. In suppl figure 1a/b, I do not understand how distance travelled was measured (and could not find it in the methods), was it always a 10min time windows at different time points?

For information about how distance travelled was measured, please refer to the methods section titled “**Behavioral Assays: Grooming**”, where we state that “locomotor activity (distance traveled, velocity) was recorded and quantified by Ethovision XT software (Noldus)” (line 553). The time points were for 10 minutes and this information can be found in the “*Behavioral Pharmacology*” section (lines 557 and 561).

3. The number of mice used is not systematically reported and sometimes difficult to spot, e.g. figure 2, suppl fig1, ..., please systematically indicates the “N=” on the figures or results section.

Thank you for pointing out this inconsistency. We have made sure to clearly define the “n” for each experiment in both the text and the figure legends throughout the manuscript.

4. In figure 3e (up), the Y-axis legend needs to be corrected to “Z-score”

Thank you for catching this typo. We have now fixed y-axis labeling as suggested.

REVIEWER COMMENTS

Reviewer #1 (Remarks to the Author):

The response to reviewers by the authors is very thorough and careful. Best of all, the authors provide new data showing that manipulation of the fronto-striatal projection regulates the effect of ketamine on compulsive grooming. However, I have three issues that still should be addressed.

1) With respect to the SAPAP3 KO mice as a model of compulsive behavior in OCD, this mouse model is frequently used, but is not well validated. SAPAP3 KO mice show a rapid response to SSRI treatment, unlike OCD patients. In sum, the SAPAP3 KO mice do show compulsive behavior, but there is no link to the compulsive behaviors in OCD patients. The term “validated” should be removed.

2) The increase in grooming over time following saline injections in SAPAP3 KO mice, is unlikely to be a real effect. Tukey post-hoc tests should only be used for the last step of analysis. ANOVAs with corrections for multiple comparisons, need to be applied until the last level of comparisons. Considering the number of bars and comparisons being made in Supp Fig 4, the two bars for SAPAP3 KO mice treated with saline at baseline vs. day 1 (with error bars that are barely nonoverlapping) are unlikely to be significantly different.

3) The images in Supplemental Figure 1 are low quality. No DAPI was used and there are no landmarks of the brain that are visible. Lines are simply drawn onto the images to show where in the brain the fluorescence is. Further, the image of the CAV2 injection site in Supplemental Figure 1 shows significant tissue damage. This can be a problem for the interpretation of the results.

Reviewer #2 (Remarks to the Author):

I appreciate the extensive and careful revisions. Indeed, the manuscript is much improved. Although not all suggestions could be implemented, the additional closed-loop optogenetic experiments and the final experiments in Fig 7 are very strong. I’m happy with the revisions and recommend publication.

Minor typo correction, line 146 “faster than grooming frequency”

Minor methods questions:

-How long did the authors wait after virus injection to perform behavior?

-How was the laser triggered for closed-loop experiments? Manually or by some automated method to detect grooming bouts?

Reviewer #3 (Remarks to the Author):

In their rebutal, the authors have done a massive amount of work including complementary analysis and experiments to adress reviewers issues. It has answered most of the concerns we could have but some questions/remarks on crucial experiments remain:

-The authors tried to confirm their original optogenetic results due to the reasonable assumption made by one of the reviewer that the effect they observed could be due to heating induced by prolonged laser stimulation. To do so, they peromed several different neuromodulation approaches. Unfortunately and very honestly, they reported that different DREADD protocols did not work in their hands and they comment their results extensively in the review manuscript. This is a concern since it could favor the idea of unspecific heating hypothesis proposed by one reviewer. Thus, they tried another approach by doing temporally-precise optogenetic inhibition/activation and found similar results. These new set of experiments could be very conclusive but I think they miss a major control with empty virus, especially due to the crucial timing of laser activation at the time of the grooming. Indeed, sudden light activation could disturb grooming behavior and even if they do not observe this effect in their inhibition protocol (although different wavelength), it would be greatly appreciated, especially since dreadd experiment were not conclusive. I understand that it may add more work but I think their study will be more convincing with this control.

-The authors used the term closed-loop optogenetic protocol but the detailed procedure is never described in the methodology other than "the laser was turned on and off with the initiation and termination of grooming respectively". I assume that the initiation/termination were done manually by the experimenter, which is fine, but should be acknowledge in the methodology.

-Since closed-loop protocol usually refer to more automated process relying on precise detection of complex biomarkers (LFP power or spiking pattern), the wording "closed-loop" sounds maladapted and

abusively employed in this case. It should be replaced by "temporally-precise optogenetic activation/inhibition" as described in their response to reviewers (or something equivalent).

-Last concern I could have that I identify would be of crucial importance is the ketamine dose used as mentioned by one of the reviewer. Since I am not an expert in that field I will not comment on that but I can understand that it could be of concern and hope that it was correctly adressed according to other reviewers.

REVIEWER COMMENTS

Reviewer #1 (Remarks to the Author):

The response to reviewers by the authors is very thorough and careful. Best of all, the authors provide new data showing that manipulation of the fronto-striatal projection regulates the effect of ketamine on compulsive grooming. However, I have three issues that still should be addressed.

1) With respect to the SAPAP3 KO mice as a model of compulsive behavior in OCD, this mouse model is frequently used, but is not well validated. SAPAP3 KO mice show a rapid response to SSRI treatment, unlike OCD patients. In sum, the SAPAP3 KO mice do show compulsive behavior, but there is no link to the compulsive behaviors in OCD patients. The term “validated” should be removed.

The term “validated” has been removed from page 4, line 40.

2) The increase in grooming over time following saline injections in SAPAP3 KO mice, is unlikely to be a real effect. Tukey post-hoc tests should only be used for the last step of analysis. ANOVAs with corrections for multiple comparisons, need to be applied until the last level of comparisons. Considering the number of bars and comparisons being made in Supp Fig 4, the two bars for SAPAP3 KO mice treated with saline at baseline vs. day 1 (with error bars that are barely nonoverlapping) are unlikely to be significantly different.

To address this important concern, we sought advice from the UCSF’s Clinical & Translational Science Institute Biostatistics Consultation Services. Our consultation was with Dr. John Boscardin, who holds a Ph.D. in biostatistics and holds an appointment as Professor of Epidemiology & Biostatistics. After consulting with Dr. Boscardin, we made several key revisions:

(1) We have confirmed with Dr. Boscardin that the Tukey method of multiple comparisons for the primary repeated measures analysis, while perhaps overly conservative, is a standard and acceptable method.

(2) Dr. Boscardin agreed with the reviewer that the analysis in the original Supplemental Figure 4 needed to be re-assessed. He recommended a simpler strategy for reporting the question of whether saline increases grooming compared to baseline at day 1 post-injection in KO mice, as described below.

- a) Following advice of Howell (https://www.uvm.edu/~statdhtx/StatPages/More_Stuff/RepMeasMultComp/RepMeasMultComp.html) and the technical documentation for PRISM (<https://www.graphpad.com/support/faq/alternative-methods-of-calculating-multiple-comparison-tests-after-repeated-measures-one-way-anova/>), we were advised to use a paired t-test to specifically address the question of whether KO-saline mice groom more 24 hours post-injection compared to baseline. This test indicated a trend toward statistical significance ($p = 0.0814$, with a mean increase in grooming of 43.62 seconds (95% confidence interval -6.549 to 93.79)). Dr. Boscardin, in agreement with the above citations and other literature (Bacchetti P. Peer review of

statistics in medical research: the other problem. *Br Med J*, 2002; 324:1271-723) did not recommend further formal adjustment for multiple comparisons in this instance.

b) We have removed Supplemental Figure 4 from the paper and have updated the corresponding text to reflect this analysis (lines 146-148, and lines 444).

3) The images in Supplemental Figure 1 are low quality. No DAPI was used and there are no landmarks of the brain that are visible. Lines are simply drawn onto the images to show where in the brain the fluorescence is. Further, the image of the CAV2 injection site in Supplemental Figure 1 shows significant tissue damage. This can be a problem for the interpretation of the results.

The images for Supplemental Figure 1 have been updated. These images were taken with a 6D Widefield Nikon Ti Inverted Microscope at the lowest magnification (4x) on the microscope, but we attempted to center the images differently to capture more macroscopic landmarks in the brain (e.g., white matter tracts). Additionally, we have provided a DAPI overlay to help with visualization of the images. The lines that are drawn on the image to demarcate PFC subregions are derived from the corresponding mouse brain atlas image that was carefully selected to match the rostro-caudal coronal slice that we imaged. In rostral dmPFC, DAPI does not always provide a clear indication of white matter tracts (which are more easily demarcated in caudal dmPFC), so below we have provided additional images to highlight these landmarks by overlaying a brightfield image. The image for the CAV2-Cre injection site has also been updated to an image that is more representative and with one that contains DAPI staining. We understand the concern about tissue damage at the injection site, though there was no apparent tissue damage to that slice when viewed under brightfield. One possible contributing factor for the patchy mCherry expression in the original image could be the anatomical subdivision of the striatum into striosome and matrix compartments, which can lend slices a swiss cheese-like appearance (where striosomes appear as black holes in the fluorescence expression). While it is impossible to insert a needle into the brain and cause absolutely no damage, we do attempt to mitigate these damages with slow insertion and removal of the needle as well as slow viral infusion rate (100 nL per minute). Additionally, any possible tissue damage from the injection process is substantially less than what is done by the optical fiber implants placed into the striatum. Finally, our control groups receive the same surgical procedures to control for any potential effects from tissue damage.

Reviewer #2 (Remarks to the Author):

I appreciate the extensive and careful revisions. Indeed, the manuscript is much improved. Although not all suggestions could be implemented, the additional closed-loop optogenetic experiments and the final experiments in Fig 7 are very strong. I'm happy with the revisions and recommendation publication.

Minor typo correction, line 146 "faster than grooming frequency"

Thank you for catching this typo; it has been fixed.

Minor methods questions:

-How long did the authors wait after virus injection to perform behavior?

The time between virus injection and behavioral testing is listed below. We have also added this information to the methods section in the lines specified in parentheses.

Fiber Photometry: 4-5 weeks (lines 610)

NpHR experiments: 7-11 weeks (lines 599, 602, 606)

ChR2 experiments: 6-9 weeks (lines 588 and 594)

-How was the laser triggered for closed-loop experiments? Manually or by some automated method to detect grooming bouts?

The laser was triggered on and off by TTL pulses sent by the Ethovision XT software (Noldus). The TTL pulse was controlled by manual keyboard strokes at the start and end of grooming. This information has now been added into the methods section for further clarification (lines 592-594).

Reviewer #3 (Remarks to the Author):

In their rebutal, the authors have done a massive amount of work including complementary analysis and experiments to adress reviewers issues. It has answered most of the concerns we could have but some questions/remarks on crucial experiments remain:

-The authors tried to confirm their original optogenetic results due to the reasonable assumption made by one of the reviewer that the effect they observed could be due to heating induced by prolonged laser stimulation. To do so, they performed several different neuromodulation approaches. Unfortunately and very honestly, they reported that different DREADD protocols did not work in their hands and they comment their results extensively in the review manuscript. This is a concern since it could favor the idea of unspecific heating hypothesis proposed by one reviewer. Thus, they tried another approach by doing temporally-precise optogenetic inhibition/activation and found similar results. These new set of experiments could be very conclusive but I think they miss a major control with empty virus, especially due to the crucial timing of laser activation at the time of the grooming.

Indeed, sudden light activation could disturb grooming behavior and even if they do not observe this effect in their inhibition protocol (although different wavelength), it would be greatly appreciated, especially since dread experiment were not conclusive. I understand that it may add more work but I think their study will be more convincing with this control.

Thank you for acknowledging that the temporally-precise optogenetic manipulations included in Figure 4f-h and Figure 5b-d address the previous concern about potential heating with continuous laser illumination. Regarding the control with an empty virus, we agree that this is indeed a critical control, and have done these experiments. We use an eYFP control group (an "empty virus" with no opsin, just fluorophore-only) showing that light alone in the absence of opsin has absolutely no effect on the animal's behavior. All of our temporally-precise optogenetic inhibition/activation experiments have eYFP virus controls and that data is shown in the figures (WT eYFP and KO eYFP). These controls demonstrate that the light produced when the laser is turned on is not disruptive to the behavior, as our eYFP controls show no differences in grooming when the laser is off versus when the laser is on (Figure 4g and Figure 5c). We have added clarifying text defining our eYFP controls for these experiments. Specifically, to the methods section we have added the following: "For both inhibition and stimulation experiments of dmPFC-DMS projections, AAV5-CaMKII α -eYFP virus, which lacks the light-responsive opsins, was used to control for surgical procedures, non-specific effects of AAV5 viral expression, and any non-opsin specific behavioral changes induced by light changes when lasers are switched on and off" (lines 521-524). Additional clarifying text can also be found in lines 244, 286, and 527.

-The authors used the term closed-loop optogenetic protocol but the detailed procedure is never described in the methodology other than "the laser was turned on and off with the initiation and termination of grooming respectively". I assume that the initiation/termination were done manually by the experimenter, which is fine, but should be acknowledge in the methodology.

The laser was triggered on and off by TTL pulses sent by the Ethovision XT software (Noldus). The TTL pulse was controlled by manual keyboard strokes at the start and end of grooming. This information has now been added into the methods section for further clarification (lines 592-594).

-Since closed-loop protocol usually refer to more automated process relying on precise detection of complex biomarkers (LFP power or spiking pattern), the wording "closed-loop" sounds maladapted and abusively employed in this case. It should be replaced by "temporally-precise optogenetic activation/inhibition" as described in their response to reviewers (or something equivalent).

Thank you for pointing out that our use of the term "closed-loop" may be misleading to some. We were trying to convey that these optogenetic experiments were distinct in that the animal's behavior determined when the laser was on and when the laser was turned off. We have now replaced the terms "open-loop" and "closed-loop" with the terms "time-dependent" and "behavior-dependent," respectively, and have adjusted the corresponding text and figure captions (lines 264, 270, 271, 277, 288, 302, 578, 584, 588, 597, 600, and 601). We believe these new terms accurately represent the difference between the two optogenetic stimulation protocols. We think that this new terminology is also clearer than saying "temporally-precise optogenetic activation/inhibition" because technically even the "time-

dependent” stimulation could be considered temporally-precise in that we are delivering defined pulses of light to trigger temporally-precise action potentials at 10Hz.

-Last concern I could have that I identify would be of crucial importance is the ketamine dose used as mentioned by one of the reviewer. Since I am not an expert in that field I will not comment on that but I can understand that it could be of concern and hope that it was correctly addressed according to other reviewers.

We would like to assure Reviewer 3 that this issue was satisfactorily addressed with this other reviewer in the last round of revisions. Supplementary Figure 11 presents the behavioral data we collected after administering a lower dose of ketamine (20mg/kg). Additionally, our main text provides citations for other papers in which the 30 mg/kg dose of ketamine was used (lines 569-572) for behavioral experiments.